



# Assimilation of SMOS Brightness Temperatures or Soil Moisture Retrievals into a Land Surface Model

Gabriëlle J. M. De Lannoy[1] and Rolf H. Reichle[2]

[1]KU Leuven, Department of Earth and Environmental Sciences, Heverlee, Belgium
[2]NASA Goddard Space Flight Center, Global Modeling and Assimilation Office, Greenbelt, Maryland

*Correspondence to:* Gabriëlle J. M. De Lannoy (gabrielle.delannoy@kuleuven.be)

**Abstract.**

Three different data products from the Soil Moisture Ocean Salinity (SMOS) mission are assimilated separately into the Goddard Earth Observing System Model, version 5 (GEOS-5) to improve estimates of surface and root-zone soil moisture. The first product consists of multi-angle, dual-polarization brightness temperature (Tb) observations at the bottom of the atmosphere extracted from Level 1 data. The second product is a derived SMOS Tb product that mimics the data at 40° incidence angle from the Soil Moisture Active Passive mission. The third product is the operational SMOS Level 2 surface soil moisture (SM) retrieval product. The assimilation system uses a spatially distributed ensemble Kalman filter (EnKF) with seasonally varying climatological bias mitigation for Tb assimilation, whereas a time-invariant cumulative density function matching is used for SM retrieval assimilation. All assimilation experiments improve the soil moisture estimates compared to model-only simulations during the period 1 July 2010 to 1 May 2015 and for 187 sites across the United States. Especially in areas where the satellite data are most sensitive to surface soil moisture, large skill improvements (e.g. increase in anomaly correlation by 0.1) are found in the surface soil moisture. The domain-average surface and root-zone skill metrics are similar among the various assimilation experiments, but large differences in skill are found locally. The observation-minus-forecast residuals and analysis increments reveal large differences in how the observations add value in the Tb and SM retrieval assimilation systems. The distinct patterns of these diagnostics in the two systems reflect observation and model errors patterns that are not well captured in the assigned EnKF error parameters. Consequently, a localized optimization of the EnKF error parameters is needed to further improve Tb or SM retrieval assimilation.

## 1 Introduction

Microwave satellite missions are collecting large amounts of data for soil moisture monitoring. It is not yet clear, however, how this wealth of data can be used in the most efficient way to obtain global estimates of soil moisture that can improve, e.g., weather prediction, flood and drought modeling, agricultural yield monitoring, or landslide predictions. Many such applications require knowledge of soil moisture in a deeper layer, where water is extracted by plant roots or stored to buffer drainage and runoff, not the approximately 5 cm surface layer to which the current L-band (∼1.4 GHz) microwave missions are sensitive. Moreover, L-band satellite observations have a fairly coarse spatial resolution (about 40 km) and are available





only at particular overpass times, typically once every 2-3 days for a given location. The challenge is thus to derive soil profile moisture information at all times and locations through data assimilation, that is, through the merger of satellite observations with information from a dynamical land surface model.

The Soil Moisture Ocean Salinity (SMOS; Kerr et al., 2010) mission and the Soil Moisture Active Passive (SMAP; Entekhabi et al.,
2014) mission are the two L-band observatories currently orbiting in space with the specific aim to measure global soil moisture. These missions supply Level 1 (L1) brightness temperature (Tb) data, Level 2 (L2) surface soil moisture (SM) retrievals and derived Level 3 (L3) products. The SMAP mission also provides an operational Level 4 Surface and Root-Zone Soil Moisture product (L4_SM; Entekhabi et al., 2014; Reichle et al., 2016) that is based on the assimilation of L1 SMAP Tb data into Goddard Earth Observing System Model, version 5 (GEOS-5) land surface simulations. Alternatively, a soil moisture
assimilation system could ingest L2 SM retrievals instead of L1 Tb observations.

In this paper, we compare Tb and SM retrieval assimilation using a historical (5-year) record of SMOS observations over North America in an assimilation system similar to that of the SMAP L4_SM system. The main differences between the SMAP L4_SM system and the experiments in this paper pertain to the differences in assimilated data, to the difference in spatial resolution of the resulting soil moisture products (36 km in the current paper, see below; 9 km for the L4_SM product), and
to differences in meteorological forcing input (re-analysis meteorology in the current paper; operational forecast meteorology corrected with gage-based precipitation in the L4_SM product).

A key disadvantage of a system that assimilates SM retrievals is that potentially inconsistent ancillary data (such as soil temperature) are used in the assimilation system and in the retrieval algorithm that generates the SM observations. However, it is more difficult to assimilate Tb observations than SM retrievals because brightness temperatures are only indirectly connected
with the land surface variables of interest and the Tb data come in multiple polarizations. SMOS Tb observations are even more complex because of their multi-angular nature. Some of the SMOS L1 Tb data complexity is reduced in the L3 SMOS Tb product and further addressed in Munoz-Sabater et al. (2014) and De Lannoy et al. (2015), who prepared the L1 SMOS Tb data for assimilation into (quasi-)operational systems. Successful examples of SMOS Tb assimilation using a variety of simplifying assumptions are illustrated in Lievens et al. (2015); De Lannoy and Reichle (2016); Kornelsen et al. (2016). These studies use
a radiative transfer model (RTM) to dynamically invert Tb information into corrections to modeled soil moisture estimates. In this paper, we advance the spatially distributed multi-angle and dual-polarization Tb assimilation of De Lannoy and Reichle (2016) in the GEOS-5 land surface model with a new version of Tb observations and an improved treatment of the observation predictions. Moreover, to mimic SMAP Tb assimilation we also assimilate dual-polarization single-angle 40° SMOS Tb observations after fitting the multi-angle Tb data (De Lannoy et al., 2015).
The current SMOS SM retrievals have been found to be skillfull (Al-Yaari et al., 2014; Fascetti et al., 2016), and research is ongoing to further improve them (Rodriguez-Fernandez et al., 2015; Ye et al., 2015; Zhao et al., 2015; van der Schalie et al., 2016; Wigneron et al., 2016). The use of these SMOS SM retrievals has been manifold, e.g. to derive enhanced estimates of precipitation (Wanders et al., 2015; Koster et al., 2016), to derive off-line root-zone soil moisture estimates (Ford et al., 2014), or to off-line downscale the data to higher-resolution soil moisture estimates (Piles et al., 2014). Other studies have assimi-
lated SMOS SM retrievals on-line into land surface models to possibly downscale the retrievals and consistently improve soil





moisture and other land surface variables (Ridler et al., 2014; Zhao et al., 2014; Lievens et al., 2015), leading to e.g. improved estimates of floods (Alvarez-Garreton et al., 2015) and crop growth (Chakrabart et al., 2014). In this paper, we use a spatially distributed assimilation system to integrate SMOS SM retrievals into the GEOS-5 land surface model with the aim to infer improved surface and root-zone soil moisture estimates. Our study mainly differs from the above SMOS SM retrieval stud-
ies in the continental and multi-year scale of the experiments, in the treatment of the SM retrieval observations, and in the comparison between Tb and SM retrieval assimilation

To assess the potential of Tb and SM retrieval assimilation, five years of SMOS Tb data or SM data are assimilated into the GEOS-5 land surface model, using a careful data quality control and data preprocessing. The observations are associated with a realistic antenna pattern, containing 50% of the signal power in a circular area with 20 km radius. Special attention is paid
to large-scale patterns of random and persistent forecast and observation errors in the different assimilation systems, and to the impact of the different assimilation schemes on the skill of surface and root-zone soil moisture estimates. Section 2 describes the SMOS observations, the various modeling components, and the in situ validation data. Section 3 highlights the technical differences between the various assimilation schemes, and section 4 presents the results.

## 2  Data and Model

### 2.1  SMOS Tb Observations

The Microwave Imaging Radiometer with Aperture Synthesis (MIRAS) onboard SMOS provides multi-angle Tb data, with a nominal (3 dB) spatial resolution of 43 km and a global coverage approximately every 3 days (at either 0600 or 1800 local time, i.e., ascending or descending half-orbits, separately). The most recent version (v620) of the SCLF1C Tb data is used. Observations are retained for further processing only (a) in the alias-free zone, (b) when the data are not contaminated by
point source radio frequency interference (RFI) or tails thereof, (c) when the values fall within the range 100-320 K, and (d) when valid data are available for both horizontal (H) and vertical (V) polarization. The flag for snapshot RFI is not activated, because it is currently too sensitive (pers. comm. R. Oliva, Y. Kerr). After the initial screening, we correct the L1 Tb values for geometric and Faraday rotation and for atmospheric and reflected extraterrestrial radiation (De Lannoy et al., 2015) using Modern-Era Retrospective Analysis for Research and Applications (MERRA) version 2 (MERRA2; Bosilovich et al., 2015)
background fields. The resulting Tb values at the bottom of the atmosphere are then binned into 41 evenly spaced angular bins with the center angle ranging from 20° through 60°. Next, the data are regridded from the 15 km Discrete Global Grid (DGG) on which they are posted to the 36-km cylindrical Equal-Area Scalable Earth (EASEv2) grid (Brodzik et al., 2014), and the data are screened for excessive sub-36-km heterogeneity (spatial standard deviation > 7 K), which is indicative of open water bodies or RFI. Tb values for a given 36-km EASEv2 grid cell are computed only if at least two valid DGG observations are
available.

From these preprocessed Tb data, two datasets are derived for assimilation: (i) a 7-angle Tb dataset, with incidence angles $\theta=[30°, 35°, 40°, 45°, 50°, 55°, 60°]$ (De Lannoy et al., 2013), and (ii) a fitted Tb dataset (De Lannoy et al., 2015) from which only the Tb at 40° incidence angle is used to mimic the single-angle nature of SMAP Tb observations. We refer to these datasets





as Tb_7ang and Tb_fit, respectively. Tb_fit data are only retained when the fitting error is less than 5 K and a minimum of 15 data points contributes to the entire fitted angular signature, with at least 5 data points above and below the 40° incidence angle and at least 10 data points in the incidence angle interval ]30°-50°[.

## 2.2 SMOS SM Retrieval Observations

The SMOS SM retrievals are extracted from the SMUDP2 product v552. Because this product version ends in early May 2015, we limit our study period to 1 July 2010 - 1 May 2015. (The reprocessed v620 version of the SM retrievals was not yet available at the time we conducted the experiments.) The SMOS retrieval algorithm simultaneously retrieves soil moisture and vegetation opacity, by fitting multi-angle Tb observations at both H- and V-polarization with simulations of the L-band Microwave Emission of the Biosphere Model (L-MEB, Wigneron et al., 2007). The SM data are retained only if: (a) all retrieved variables

fall within a realistic range (0-0.6 $m^3.m^{-3}$ for soil moisture), (b) the SM uncertainty estimated by the SMOS retrieval algorithm is less than 0.1 $m^3.m^{-3}$, (c) the RFI probability for both H- and V-polarization is less than 0.3, and (d) SM retrieval flags are not raised for high topographic complexity, high urban fraction, high open water fraction, sea ice, coastal areas, and high total electron content. Further screening for frozen temperature and snow is based on GEOS-5 model output (section 2.3). After the regridding from the 15-km DGG grid to the 36-km cylindrical EASEv2 grid, the data are screened for excessive sub-36-km

heterogeneity (spatial standard deviation $> 0.2$ $m^3.m^{-3}$). SM values for a given 36-km EASEv2 grid cell are computed only if at least two valid DGG observations are available.

## 2.3 Soil Moisture and Brightness Temperature Modeling

The land data assimilation system used here employs the GEOS-5 Catchment land surface model (CLSM; Koster et al., 2000), along with an L-band tau-omega radiative transfer model (RTM; De Lannoy et al., 2013, 2014b). The CLSM simulations use

GEOS-5 parameters (Mahanama et al., 2015; De Lannoy et al., 2014a) similar to those used in the SMAP L4_SM product, and are forced with $1/2°\times2/3°$ GEOS-5 forcing data from MERRA (Rienecker et al., 2011) bilinearly interpolated to the model grid. The study domain covers most of North America, with the northwestern corner at (125°W, 55°N) and the southeastern corner at (60°W, 24°N).

    The computational elements are the 36-km EASEv2 grid cells. The land model computation time step is 7.5 minutes, and

output is saved at 3-hourly intervals. At each grid cell, the surface soil moisture content (sfmc, 0-5 cm) and root-zone soil moisture content (rzmc, 0-100 cm) are diagnosed based on three prognostic variables: catchment deficit (catdef), root-zone excess (rzexc), and surface excess (srfexc). Similarly, the surface (skin) temperature is diagnosed from the prognostic land surface temperatures across the saturated (tc1), unsaturated (tc2), and wilting (tc4) sub-grid areas. Finally, the soil temperature (tp1 for the topmost layer) is diagnosed from the prognostic ground heat content (ght1 for the top layer). An overview of the

model variables is given in Reichle et al. (2015).

    The L-band tau-omega RTM converts the 36-km CLSM soil moisture and temperature simulations into 36-km L-band Tb estimates when the soil is not frozen or covered with snow, when precipitation is less than 10 mm/day, and where the open water fraction is less than 5% percent. For each 36-km grid cell, key parameters of the RTM are estimated by minimizing Eq



B.1 in De Lannoy et al. (2014b), using a 5-year history of SMOS v620 Tb data, and computing observation predictions (see below) at the footprint scale. Specifically, all 36-km grid cells within one footprint area are assigned the same set of RTM parameters, while the dynamic background information is spatially variable. The calibration estimates the RTM parameters for the entire footprint area and the resulting values are assigned to the central 36-km grid cell only. The RTM is calibrated using all 5 years of available Tb data and aims at minimizing climatological biases. The data assimilation is performed over the same 5 years and aims at addressing random (or short term) errors.

For the computation of differences between SMOS observations and footprint-scale model simulations in the RTM calibration and for the computation of the "observation-minus-forecast" (O-F) residuals in the assimilation system (section 3.1), the modeled 36-km soil moisture or Tb simulations are aggregated to the footprint scale by spatial convolution with weights given by an approximation of the SMOS antenna pattern. We also refer to these spatially aggregated model estimates as 'observation-tion predictions'. The SMOS antenna pattern is approximated by a two-dimensional Gaussian function containing 50% of the signal within a circle with a radius of 20 km. The simulations outside a radius of 40 km are discarded in the computation of the footprint-scale estimates. The number of 36-km EASEv2 grid cells included in one footprint area varies with latitude. The circular footprint shape is preserved everywhere on the globe. In contrast, the shape of the EASEv2 grid cells projected on the globe varies with the latitude, with an aspect ratio of 1 at 30° (north/south) latitude, larger than 1 towards the poles and less than 1 towards the equator. Therefore, at higher latitudes multiple EASEv2 grid cells with the same latitude and various longitudes belong to one circular footprint, whereas towards the equator, several EASEv2 grid cells with the same longitude and various latitudes contribute to the footprint. Overall, the difference between single 36-km simulations and footprint-scale values is small, but the number of valid Tb observation predictions at the footprint scale is reduced, because of the increased likelihood of finding a 36-km grid cell with a non-negligible water fraction, snow amount, or precipitation, within the footprint area.

## 2.4 In Situ Soil Moisture Data

The assimilation results are evaluated using independent in situ measurements of surface and root-zone soil moisture from two sparse networks across the US: the US Natural Resources Conservation Service Soil Climate Analysis Network (SCAN; Schaefer et al., 2007) and the US Climate Reference Network (USCRN; Diamond et al., 2013; Bell et al., 2013). Surface soil moisture measurements are taken at approximately 5 cm depth. Root-zone soil moisture measurements are a weighted average of measurements at 5, 10, 20, and 50 cm depth. Given the difference in spatial support between these point measurements and the 36-km gridded model and assimilation results, the skill is quantified in terms of anomaly time series correlation (anomR), and unbiased root-mean-square difference ($\mathrm{RMSD}_{ub}$; Entekhabi et al., 2010), using all 3-hourly forecast and analysis time steps in the period 1 July 2010-1 May 2015, excluding times when the soil is frozen or snow covered. Metrics at a single site are only calculated if at least 200 data points are available. Skill metrics across an entire network are calculated by clustering the sites within SCAN and USCRN to avoid that densely sampled areas dominate the validation metrics and to ensure realistic confidence intervals (De Lannoy and Reichle, 2016). The number of clusters is estimated a priori after prescribing an average





cluster radius of $3°$, which approximately reflects the autocorrelation length of large-scale topographic and meteorological phenomena. The actual size of the clusters that results from the clustering algorithm varies strongly in space.

## 3 Data Assimilation

### 3.1 Distributed Ensemble Kalman Filter

For both Tb and SM retrieval assimilation, a spatially distributed (or three-dimensional, 3D) ensemble Kalman filter (EnKF; Reichle and Koster, 2003; De Lannoy and Reichle, 2016) is used. This system simultaneously assimilates multiple spatially distributed observation sets to update the simulations at each 36-km model grid cell. The details of the Tb assimilation system are explained in De Lannoy and Reichle (2016) and differ only in that the observations are here associated with a spatially variable antenna pattern reaching out to a radius of 40 km.

During the model integration, a data assimilation step is activated every 3 hours. All the SMOS observations $\mathbf{y}_i$ collected within 1.5 hours of the analysis time $i$ are assimilated simultaneously to update the forecasted state $\hat{\mathbf{x}}_{k,i}^{j-}$ at location $k$ as follows:

$$\hat{\mathbf{x}}_{k,i}^{j+} = \hat{\mathbf{x}}_{k,i}^{j-} + \mathbf{K}_{k,i}[\mathbf{y}_i^j - \hat{\mathbf{y}}_i^{j-}].$$
(1)

with $j$ denoting the ensemble member, $\mathbf{K}_{k,i}$ the Kalman gain, $\mathbf{y}_i^j$ the perturbed observations, $\hat{\mathbf{y}}_i^{j-} = \mathbf{h}_i(\hat{\mathbf{x}}_i^{j-})$ the observation predictions, and $\mathbf{h}_i(.)$ the observation operator mapping the simulated land surface variables to observed quantities. Bias in the

observation-minus-forecast residuals is addressed prior to the analysis (section 3.2). The ensemble is created by perturbing the model forcing, the model forecasts and the observations (section 3.3). The Kalman gain is calculated as:

$$\mathbf{K}_{k,i} = \mathrm{Cov}(\hat{\mathbf{x}}_{k,i}^-, \hat{\mathbf{y}}_i^-) \left[ \mathrm{Cov}(\hat{\mathbf{y}}_i^-, \hat{\mathbf{y}}_i^-) + \mathbf{R}_i \right]^{-1},$$
(2)

where $\mathrm{Cov}(\hat{\mathbf{x}}_{k,i}^-, \hat{\mathbf{y}}_i^-)$ is the (sample) error covariance (across the ensemble) between the forecasted land surface state and the forecasted Tb or SM. Similarly, $\mathrm{Cov}(\hat{\mathbf{y}}_i^-, \hat{\mathbf{y}}_i^-)$ is the (sample) error covariance of the Tb or SM forecasts, and $\mathbf{R}_i$ is the Tb or

SM observation error covariance. The Kalman gain is identical for all ensemble members.

In the case of SM retrieval assimilation, the observation operator $\mathbf{h}_i(.)$ performs the spatial aggregation of soil moisture simulations from the 36-km grid cells to the satellite footprint; in the case of Tb data assimilation, the observation operator includes both the RTM and the spatial aggregation of gridded Tb simulations to the footprint (section 2.3). For the Tb_7ang assimilation, one observation set at location $\kappa$ contains Tb observations at a maximum of 7 angles and both H- and V-polarization, i.e., up

to 14 individual observations $y_{\lambda,\kappa,i} \in \mathbf{y}_{\kappa,i}$. The subscript $\lambda$ refers to the polarization and incidence angle of the individual Tb observations. In the middle part of the swath, all 14 observations are typically available, whereas slightly fewer observations are available in the outer portions of the swath, where the observations with lower incidence angles are missing.

For the Tb_fit assimilation, one observation set usually contains 2 observations, i.e. both H- and V-polarization Tb at $40°$ incidence angle. For the SM retrieval assimilation, each observation set contains only one observation. In all cases, the obser-

vation vector $\mathbf{y}_i^j$ collects multiple perturbed observation sets that are spatially distributed within an influence radius of $1.25°$ around the model grid cell $k$, and each observation vector $\mathbf{y}_i^j$ has a forecasted counterpart $\hat{\mathbf{y}}_i^{j-}$. After removal of the persistent



errors (section 3.2) from the O-F residuals (or innovations), the increments $\mathbf{K}_{k,i}[\mathbf{y}_i^j - \hat{\mathbf{y}}_i^{j-}]$ are calculated and applied to the state variables.

The subset of prognostic variables updated in Eq. 1 differs depending on the assimilation experiment. The state vector for Tb assimilation ($\mathbf{x} = $ [catdef, srfexc, rzexc, tc1, tc2, tc4, ght1]$^T$) includes prognostic variables related to soil moisture and soil temperature (section 2.3). In contrast, the state vector for SM retrieval assimilation ($\mathbf{x} = $ [catdef, srfexc, rzexc]$^T$) contains only model prognostic variables related to soil moisture.

Figure 1 and Figure 2 illustrate the concept for Tb assimilation and SM retrieval assimilation, respectively. Figures 1a-b show swaths of footprint-scale bias-corrected Tb_fit innovations (mapped onto the 36-km EASEv2 grid), for H- and V-polarization at 40° incidence angle from the single-angle Tb assimilation system. The Tb innovations are then transformed into soil moisture and temperature increments using Eq. 1. Where Tb innovations are warm, the soil water is reduced and the temperature is increased. For simplicity, Figure 1c shows the total profile water increments ($\Delta$wtot=$\Delta$srfexc+$\Delta$rzexc-$\Delta$catdef) and Figure 1d shows increments to the first soil layer temperature ($\Delta$tp1). Increments to the surface temperature prognostic variables (section 2.3; $\Delta$tc1, $\Delta$tc2, $\Delta$tc4) are similar (not shown). Finally, the increments are added to the forecasted fields to create spatially complete analysis maps of surface and root-zone soil moisture, as well as surface and soil temperature (Figures 1e-g).

Similarly, Figure 2a shows the SM innovations from the SM retrieval assimilation at the same time as in Figure 1. Areas with positive (wet) SM innovations in the SM retrieval assimilation roughly correspond with negative (cold) Tb innovations in the Tb assimilation system (Figures 1a-b ). Note that the colorbars for Tb and SM throughout the manuscript are chosen according to the rule of thumb that a 2-3 K change in Tb corresponds to a 0.01 m$^3$.m$^{-3}$ change in soil moisture, but keep in mind that the relationship between Tb and SM is non-linear and varies with time, location and incidence angle. Next, the SM innovations are converted to soil moisture increments ($\Delta$wtot; Figure 2b); no increment to surface or soil temperature is calculated. Figures1c and 2b show that the Tb and SM retrieval assimilation systems produce wtot increments with somewhat different large-scale patterns, which is further discussed in section 4.2. Finally, Figures 2c-d show the resulting surface and root zone soil moisture analysis fields obtained by adding the increments to the model forecast fields. For both the Tb and SM retrieval assimilation systems, the analysis increments blend smoothly into the forecast fields, that is, the analysis maps do not reveal sharp spatial edges that would reveal the geometry of the assimilated satellite swaths. Further details about this figure are discussed in section 4.1.

## 3.2 Tb and SM Innovation Bias

To limit the long-term biases between Tb observations and simulations, the RTM was calibrated (section 2.3). The 5-year average absolute bias between SMOS Tb and forecasted Tb is about 2 K across the domain. In general, slightly warm model biases are found in the boreal zones and cold model biases over the central part of the US (not shown). But larger seasonal Tb biases remain, primarily due to systematic errors in the modeled temperature and vegetation. The seasonally varying climatological Tb bias is removed prior to data assimilation for each angle, polarization and overpass time separately, as described





in De Lannoy and Reichle (2016). The Tb innovation biases are calculated over the period 1 July 2010 - 1 May 2015 for each individual 36-km grid cell without spatial sampling.

The CLSM soil moisture was not calibrated for lack of global observations that would support such an effort and because modeled soil moisture does not necessarily represent soil moisture as observed in the field anyway (Koster et al., 2009). Unlike biases in Tb innovations, the biases in the SM innovations are more stationary and do not depend on seasonal temperature variations. Therefore, the SM innovation biases are not corrected seasonally, but instead cumulative distribution function (CDF) matching between the observations and simulations is performed (Reichle et al., 2004) to reconcile the differences in long-term mean, variance and higher moments, as in earlier retrieval assimilation studies (Liu et al., 2011; Draper et al., 2012). In line with the Tb innovation biases, the SM innovation biases are computed for 1 July 2010 - 1 May 2015 at each 36-km grid cell individually.

### 3.3 Random Forecast and Observation Error

The imposed ensemble forecast perturbations for Tb and SM retrieval assimilation are identical to those of De Lannoy and Reichle (2016) and not repeated here. The total observation error standard deviation for SMOS Tb_7ang is set to 6 K, which yields near-optimal assimilation diagnostics on average across the globe. However, the diagnostics are not necessarily near-optimal in individual regions (De Lannoy and Reichle, 2016). The input observation error standard deviation for SM retrievals is 0.04 $m^3.m^{-3}$, in line with the soil moisture accuracy requirement for the recent SMOS and SMAP missions. The SM retrieval error standard deviation is rescaled following the CDF-matching of the SM observations and results in an effective mean error standard deviation of 0.02 $m^3.m^{-3}$, with larger values in the wetter eastern part, which exhibits a higher temporal variability in soil moisture simulations, and lower values in the drier, western part of the study domain (not shown). In all cases, the spatial observation error correlation length is 0.25°. In case of multi-angle Tb_7ang assimilation, interangular error correlations are imposed as in De Lannoy and Reichle (2016).

Observation errors in Tb data or SM retrievals are a combination of instrument error and representation error (Cohn, 1997; van Leeuwen, 2015). The 6 K Tb error consists of radiometric error of about 4 K for individual incidence angles (instrument error), plus 4.5 K representation inaccuracies (in our system, i.e. based on the near-optimal 6 K observation error) due to errors in the RTM, or other discrepancies between Tb observations and forecasts ($6=\sqrt{4^2+4.5^2}$). For Tb_fit observations, the instrument error may be slightly reduced compared to that for Tb_7ang after the angular smoothing, but the representation error remains similar. SM observations contain retrieval errors due to errors in the RTM and in the input L1 Tb observations, as well as representation error due to, e.g., the inherently different nature of simulated and observed soil moisture (Koster et al., 2009). In either case, the representation error depends on the soil moisture and temperature dynamics and should ideally be modeled as function of time and location, but we chose a constant input observation error standard deviation in this paper for simplicity. For SM retrieval assimilation, some spatial error variability is introduced after rescaling in line with the CDF-matching.





## 3.4 Tb or SM Retrieval Assimilation

In our experiments, we do not expect the SMOS Tb and SM retrieval assimilation systems to yield the same results. During the SMOS L2 SM retrieval optimization, the Tb data are used to estimate surface soil moisture and *vegetation opacity*, given soil temperature background fields provided by the European Center for Medium-Range Weather Forecasts (ECMWF), and

look-up parameter information that differs significantly from the NASA GEOS-5 land data assimilation system. In contrast, our SMOS Tb assimilation scheme estimates soil moisture and *temperature*, given vegetation information. Furthermore, the data screening is necessarily different for Tb data and SM retrievals, and the approach for bias correction is intentionally different. The soil moisture information extracted during the L2 retrieval process or Tb assimilation is thus by design expected to be different. Finally, differences in the Tb and SM retrieval assimilation results could also be due to differences in how close each

of the systems is to an optimal calibration of its model and observation error parameters.

## 4 Results

### 4.1 Observation and Forecast Diagnostics

#### 4.1.1 Number of Assimilated Observations

Let us revisit Figures 1a-b and 2a to further highlight some differences between the various assimilated SMOS observations.

First, the swath width for Tb innovations is much narrower than that of the SM innovations because the assimilated Tb observations are strictly limited to the alias-free zone within the full swath while the assimilated SM retrievals are retained in the extended alias-free zone. Furthermore, the swath width of the Tb_fit innovations is narrower than that of the multi-angle assimilation (not shown) because the fitting requires sufficient data at a range of incidence angles and lower angle data are not available at the outer edges of the swaths. Note that SMAP provides useable Tb measurements over a much wider swath (not

shown).

The different swath widths result in different numbers of observation sets assimilated in each of the three experiments. Figures 3a-c show the average number of assimilated observation sets (defined in section 3.1) over the study period 1 July 2010 - 1 May 2015. The number of observation sets is smallest (one every 4 days) for Tb_fit and largest for SM retrievals (one every 2 days), because the swath width is narrowest for Tb_fit and widest for SM retrievals. The northern areas and the western

mountain ranges have fewest observations, because data are not used when the soil is frozen or snow covered. Tb observations are not assimilated in many small areas scattered around the study domain where more than 5% of open water is found in the footprint, based on the underlying GEOS-5 land mask. For the SM retrievals, the screening for excessive ($> 5\%$) water fraction is only based on the product science flags, not on GEOS-5 information. Data gaps in the SM retrievals are found in the western mountain ranges and in the vegetated southeastern part of the US. The data coverage is also different for Tb and SM retrieval

assimilation because the availability of the climatological information needed for the innovation bias correction (sections 3.2) is different for the Tb and SM retrieval observations.



### 4.1.2 Actual Observation and Forecast Errors

The long-term mean observation-minus-forecast differences (O-F, or innovations) are unbiased by design (section 3.2). The Hovmüller plots for two data assimilation cases in Figure 4 reveal that the temporal pattern in area-averaged biases is fairly random for the Tb_7ang assimilation case (very similar for Tb_fit assimilation, not shown), whereas it shows a slight seasonal pattern in the SM retrieval assimilation case. This small difference is not surprising, given that the Tb innovation bias is seasonally corrected, whereas the SM innovation bias is not.

The time series standard deviation of the innovations, that is, the root-mean-square-difference (RMSD) between SMOS observations and simulations, represents the total observation and forecast error that is present in the assimilation system (Desroziers et al., 2005). The spatial patterns of this diagnostic are very different for Tb and SM retrieval assimilation. Figures 3d-e show values of about 7.4 K for Tb_7ang and Tb_fit, with larger values (exceeding 10 K) in the central plains and along the Mississippi, where agricultural practices, such as altering crop rotation and irrigation, are observed by SMOS but not simulated in the model. Along the East coast and in the Southeast, the temporal standard deviation in the innovations is low (2-3 K): forests show a limited interannual variability, and under dense vegetation Tb is only marginally sensitive soil moisture and depends primarily on vegetation characteristics and (physical) temperature.

The standard deviation in the SM innovations in the SM retrieval assimilation (Figure 3f) is 0.03 $m^3.m^{-3}$, showing larger values in the wetter vegetated East and smaller values in the drier West, with the exception of the West coast. Surprisingly, even though altering crop rotation and irrigation are not simulated, the values over the central agricultural area are not higher than elsewhere in the domain. This good agreement between SMOS SM retrievals and our simulations is partly due to the bounded nature of SM (unlike Tb) and the CDF-matching between both.

Our current system has a Tb sensitivity to soil moisture of about 1.3 K/0.01 $m^3.m^{-3}$ across the domain, averaged over all incidence angles and polarizations. A standard deviation in SM innovations of 0.03 $m^3.m^{-3}$ would thus roughly correspond to a standard deviation in Tb innovations of about 4 K, but instead we find 7.4 K across the study domain in the Tb assimilation systems. The Tb observations thus either have a comparably higher observation (incl. representation) error or they contain more information than the SM retrievals. At this point, we anticipate that the higher Tb innovations in the central plains may indicate that the Tb observations contain more unfiltered information about soil moisture (e.g. irrigation) and that the Tb observation error is higher due to shortcomings e.g. in the vegetation modeling (representation error).

### 4.1.3 Actual versus Simulated Observation and Forecast Errors

In a near-optimal filtering system, that is, a system that correctly simulates the actual model and observation errors, the standard deviation of the *normalized* innovations $[\mathbf{y}_{\kappa,i} - \hat{\mathbf{y}}^-_{\kappa,i}]_\lambda / \sqrt{[\mathbf{R}_{\kappa,i} + \mathrm{Cov}(\hat{\mathbf{y}}^-_{\kappa,i}, \hat{\mathbf{y}}^-_{\kappa,i})]_{\lambda\lambda}}$ is close to unity (Reichle et al., 2002). Figures 3g-i show that, averaged across the domain (and across all angles and polarizations for Tb assimilation), this metric is 1.14, 1.11 and 1.23 [-] for Tb_7ang, Tb_fit and SM retrieval assimilation. The figure thus suggests that, on average, the simulated errors in the assimilation system only slightly underestimate the actual errors. But the figures also show that the metric varies strongly across the domain and exhibits very different spatial patterns for Tb and SM retrieval assimilation. For





Tb_7ang and Tb_fit assimilation, values are much larger than 1 in the central area and much smaller than 1 in the eastern forested area. This indicates that the assigned observation and forecast errors are severely underestimated in the central area and overestimated in the eastern forested area. Over forests, it can be assumed that the assigned representation error (part of observation error) should be smaller. The Tb forecast error is already very small (see below), because the Tb uncertainty is

only marginally sensitive to soil moisture uncertainties under dense vegetation. For SM retrieval assimilation, the pattern is reversed, with the largest values in the eastern half of the domain, suggesting that here the simulated errors underestimate the actual errors. Values less than 1 are found in most of the western half of the domain, where the SM retrieval assimilation seems to overestimate the actual errors.

To further interpret the actual and simulated error magnitudes, Figures 3j-k show the ensemble spread in the Tb forecasts

(that is, the simulated forecast error standard deviation) $\sqrt{[\mathrm{Cov}(\hat{\mathbf{y}}_{\kappa,i}^{-}, \hat{\mathbf{y}}_{\kappa,i}^{-})]_{\lambda\lambda}}$. Averaged across all angles and polarizations $\lambda$, the values are around 2 K when averaged across the entire domain. Larger values (3 K) are found in the central and dry western part, and smaller values (1 K) in the wetter eastern part. This pattern is similar for the SM ensemble spread in the SM retrieval assimilation system (Figure 3l). In dry climates, the root-zone soil moisture often drops to the wilting point, remains stagnant and no longer replenishes the surface. This results in increased sensitivity of the surface soil moisture to perturbations

in meteorological conditions, and thus in higher uncertainty estimates for surface soil moisture in dry climates.

Given that the Tb observation error $\sqrt{[\mathbf{R}_{\kappa,i}]_{\lambda\lambda}}$ is set to 6 K for each individual angle, polarization and overpass time in the Tb assimilation, the approximate total assigned observation and forecast error is 6.1 K ($\sqrt{6^2 + 2^2}$) across the study domain, 6.7 K ($\sqrt{6^2 + 3^2}$) in the central area, and 6 K ($\sqrt{6^2 + 1^2}$) in the eastern Appalachian area. Because the assigned observation error is uniformly set to 6 K, the spatial variability in the total simulated errors is thus too small compared to the actual errors

(Figures 3d-e), which ranges from more than 10 K in the central area to and around 2-3 K in the eastern Appalachian area.

The SM observation error (after rescaling) is 0.02 m$^3$.m$^{-3}$ on average across the domain, with higher values in the eastern part and lower values in the western part, with the exception of Mexico, California and West Oregon where higher observation errors are found (section 3.3). This general pattern is reversed in the SM forecast errors. Combined, the spatial variability in the SM observation and forecast errors is not capturing the spatial variability in the actual errors (Figure 3f), which leads to an

overestimation of the errors in the West and an underestimation in the East.

## 4.2   Analysis Increments

### 4.2.1   Spatio-Temporal Patterns

The Kalman filter translates footprint-scale innovations into 36-km increments. Because of the spatially distributed (3D) filtering (section 3.1), the number of increments in Figures 5a-c is about 1.4 times the number of assimilated observation sets

(Figures 3a-c). Many areas with missing observations (or observation predictions) are filled through interpolation and extrapolation. With SM retrieval assimilation, there is almost one increment per day.

Figures 5d-f show the temporal standard deviations in the increments for the total soil profile water ($\Delta$wtot=$\Delta$srfexc+$\Delta$rzexc-$\Delta$catdef). The area average ($\pm$standard deviation) values are 6.9$\pm$3.7 mm for Tb_7ang assimilation, 5.9$\pm$3.5 mm for Tb_fit





assimilation and 4.2±1.9 mm for SM retrieval assimilation. After scaling for the (variable) profile depth, the area-average values in volumetric soil moisture units are $3.4 \pm 1.7 \times 10^{-3}$ m$^3$.m$^{-3}$ for Tb_7ang assimilation, $2.9 \pm 1.7 \times 10^{-3}$ m$^3$.m$^{-3}$ for Tb_fit assimilation and $2.3 \pm 1.9 \times 10^{-3}$ m$^3$.m$^{-3}$ for SM retrieval assimilation.

The individual components of the wtot increments are shown in Figures 5g-i for the surface excess increments, Figures 5j-l for the root zone excess increments, and Figures 5m-o for the catchment deficit increments. The patterns in wtot increments are dominated by catdef increments, and they generally reflect the patterns in the respective innovations standard deviations (Figures 3d-f), which are very different for Tb and SM retrieval assimilation. The catdef increments pertain to the entire profile depth (which varies, with typical values around 2-3 m) and have a relatively small impact on the upper 5 cm soil layer (surface soil moisture): the domain-averaged magnitude of 5.4 mm, 4.9 mm and 3.5 mm for catdef increments due to Tb_7ang, Tb_fit or SM retrieval assimilation, respectively (Figures 5m-o), would scale to about 0.1 mm for a 5 cm soil layer. This is less than the 0.6, 0.4 and 0.4 mm for the corresponding srfexc increments (Figures 5g-i), which are directly applied to the upper 5 cm soil layer. The increments in rzexc (Figures 5j-l) are relatively smallest, because this variable is not perturbed by design.

Both Tb and SM retrieval assimilation show similar spatial patterns in the standard deviations of srfexc increments (Figures 5g-i): the largest increments are found in the dry West and the smallest in the wetter East. The patterns in srfexc increments agree with the patterns in the ensemble forecast uncertainty for this variable (not shown, but implied by the Tb and soil moisture uncertainty in Figures 3j-l). The srfexc values are small with small uncertainties, and the increments are thus similarly bounded in both Tb and SM retrieval assimilation, yielding comparable spatial increment patterns.

Finally, Figure 6 compares spatially and temporally collocated wtot, srfexc and rzexc increments obtained with Tb_7ang assimilation, Tb_fit assimilation and SM retrieval assimilation, i.e., the figure shows all pairs of increments available from two assimilation cases. The scatter plots show that the increments are usually small and unbiased. The correlation between the wtot increments (Figure 6a) obtained by Tb_7ang and Tb_fit assimilation is 0.7, and aligns with the expectation that either Tb assimilation experiment roughly corrects for the same events. In contrast, the correlation between the increments obtained by Tb_7ang and SM retrieval assimilation is only 0.3 (Figure 6b). The figure is similar when comparing the Tb_fit and SM retrieval assimilation (not shown). For srfexc and rzexc (Figures 6c-f), the increments are again similar for Tb_7ang and Tb_fit assimilation, but different for Tb and SM retrieval assimilation. For all soil moisture prognostic variables, Tb assimilation leads to larger increments than SM retrieval assimilation. The different assimilation systems thus introduce distinct corrections to the modeled soil moisture trajectories.

### 4.2.2 Discussion

In a nutshell, Eq. 1 expresses that the increments are given by the product of the Kalman gain and the innovations. To explain the differences in increment patterns between Tb and SM retrieval assimilation, we must therefore consider each system's innovations and Kalman gains. The relatively larger magnitude of the Tb innovations compared to the SM innovations (section 4.1.2) contributes to the fact that the Tb assimilation results in larger soil moisture increments. This is the case even though the SM retrieval assimilation (unlike Tb assimilation) applies increments only to moisture variables and does not adjust modeled temperatures.



Furthermore, the Kalman gain matrices $\mathbf{K}_{k,i}$ (Eq. 2) for Tb and SM retrieval assimilation are different because the two systems employ different observation operators $\mathbf{h}_i(.)$ and different observation error covariances $\mathbf{R}_i$. First, we note that the non-linear inversion of Tb innovations to soil moisture increments, driven by the RTM in the observation operator, is *not* responsible for the larger wtot increments in the central grass and crop areas, because these areas exhibit low values for the microwave roughness parameter ($h <$0.2, not shown) and a high sensitivity of Tb to soil moisture (as confirmed by the high forecast Tb errors in Figures 3j-k). That is, in these areas commensurately large Tb O-F values result only in *small* updates to soil moisture.

Second, the choice of a spatially uniform observation error covariance in the Tb assimilation experiment creates an imprint of the innovation pattern in the increment pattern. Higher increments are found in the agricultural areas with large Tb innovation standard deviations (Figures 3d-e), because irrigation is not modeled and vegetation is not accurately parameterized. Since the filter is not set up to correct the latter, occasional excessive increments to soil moisture and temperature may be introduced. Such shortcomings could be mitigated by a more sophisticated assignment of Tb observation (representation) errors.

For SM retrieval assimilation, the pattern of the SM innovation standard deviation (RMSD) is similarly visible in the increments, with smaller values in the West and higher values in the East. Here again, the true spatio-temporal nature of the observation errors is not captured in the assigned observation error covariance and therefore propagated into the increments. Note also that the 0.03 $\mathrm{m^3.m^{-3}}$ SM innovation standard deviation (top 5 cm, Figure 3f) is translated to a standard deviation of profile moisture increments of 0.002 $\mathrm{m^3.m^{-3}}$ (Figures 5f rescaled by profile depth), but these increments are not equally distributed, i.e. larger increments are found for surface soil moisture and smaller increments for the deeper profile.

### 4.3  In Situ Validation

The above discussion highlights similarities and stark contrasts in how the Tb and SM retrieval assimilation systems operate. In this section, we look at the effect of these differences on the skill of the assimilation estimates versus in situ observations. Figure 7 shows the change in $\mathrm{RMSD}_{ub}$ (section 2.4) between the model-only open loop (OL) simulation and either the Tb_7ang or SM retrieval data assimilation (DA) experiment ($\Delta\mathrm{RMSD}_{ub} = \mathrm{RMSD}_{ub}$(DA) - $\mathrm{RMSD}_{ub}$(OL)) at individual SCAN and USCRN sites, for the period 1 July 2010 - 1 May 2015. The green background shading indicates areas with modest topographic complexity and vegetation cover and where the satellite observations are most sensitive to surface soil moisture (details in De Lannoy and Reichle, 2016). On average, both assimilation experiments introduce improvements at about 80% of the sites for surface soil moisture, with spatially averaged $\Delta\mathrm{RMSD}_{ub}$ values of -0.004 and -0.003 $\mathrm{m^3/m^{-3}}$ for Tb_7ang and SM retrieval assimilation, respectively. (Spatial average metrics are computed using a cluster-based algorithm, Section 2.4.) The improvements are also propagated to the root-zone soil moisture (65% of sites improved) with smaller average $\Delta\mathrm{RMSD}_{ub}$ values of -0.002 and -0.001 $\mathrm{m^3/m^{-3}}$, respectively. Both Tb and SM retrieval assimilation show improvements in the central and eastern parts of the US but perform poorly in the western mountain areas. The Tb_7ang assimilation shows the largest improvements in the central US, whereas the SM retrieval assimilation shows the largest improvements in the southeastern part, for both surface and root-zone soil moisture. It is possible that the Tb assimilation has a larger impact in the central US than the SM retrieval assimilation, because irrigation events may be filtered in the SM retrievals (and perhaps partly assigned





to vegetation opacity retrievals). As will be shown next, the domain-average skill differences between Tb_7ang, Tb_fit or SM retrieval assimilation are not significant.

The barplots in Figure 8 summarize the average anomR values for the open loop and data assimilation experiments, after stratifying all SCAN and USCRN sites into 'favorable' and 'non-favorable' categories, where the 'favorable' sites are located
the area where the satellite observations are most sensitive to soil moisture (indicated with green background shading in Figure 7). The figure shows that the open loop anomR values for surface soil moisture are similar for both the favorable and non-favorable areas (0.51 and 0.50, respectively). However, data assimilation has a larger impact in favorable areas where all assimilation schemes introduce significant improvements (anomR=0.63, 0.61 and 0.59 for Tb_7ang, Tb_fit and SM retrieval assimilation). In non-favorable areas, the improvements are smaller but still significant (anomR=0.57, 0.56 and 0.54, for
Tb_7ang, Tb_fit and SM retrieval assimilation).

In the root-zone, data assimilation also improves the skill over the open loop simulations, but without statistical significance. The open loop simulations yield anomR values of 0.56 and 0.50 in favorable and non-favorable areas, respectively. In favorable areas, the assimilation increases the anomR to 0.64, 0.64 and 0.62, for Tb_7ang, Tb_fit and SM retrieval assimilation. In non-favorable areas, the skill improvement is limited and the anomR values are 0.54, 0.54 and 0.52, for Tb_7ang, Tb_fit and SM
retrieval assimilation. In any case, with assimilation, all anomR values exceed 0.5, meaning that the skill becomes better than a climatological forecast (Brier skill score larger than 0).

Overall, the skill metrics are comparable for the Tb_7ang and Tb_fit assimilation (Figure 8). The results from SM retrieval assimilation are slightly worse than those from Tb assimilation, which may indicate that Tb observations indeed still contain more information (Section 4.2) than the SM retrievals, which are implicitly filtered during the retrieval process. Yet, the
differences between the domain-averaged skill values of the various assimilation schemes are minimal. Furthermore, when running the assimilation scheme with different spatially constant Tb observation error parameters, the skill metrics only changed marginally. This reveals that our skill metrics are relatively insensitive to uniform changes in the data assimilation parameters. One reason for this is that the skill metrics are presented as (clustered) spatial averages, which compensate for large local differences. It is expected that the skill of our data assimilation systems can only be further improved by using a more localized (in
space and time) approach to optimizing the assimilated observations (e.g. L2 SM retrievals), and the forecast and observation error parameters in the EnKF.

Finally, unlike Liu et al. (2011), the skill improvements in this study are smaller when we correct the re-analysis precipitation input with gauge-based precipitation data (Reichle and Liu, 2014). This and other recent improvements in the GEOS-5 modeling system make it increasingly challenging to obtain significant skill improvements from the assimilation of microwave
observations over areas for which high-quality forcing data are available, such as the domain studied here. The benefits of the microwave-based soil moisture assimilation system are expected to be greater in areas with poorer ancillary inputs to the modeling system. This aspect will be further investigated through the validation of the global SMAP L4_SM data product.



## 5   Conclusions

The SMOS and SMAP satellite missions currently provide a wealth of L-band data to monitor large-scale soil moisture. A key question is how to make the best use of these data in current land surface data assimilation systems. The L1 Tb data from these missions are often complex, because of their multi-polarization and possibly multi-angle nature and their indirect connection

with soil moisture. In theory, the best approach is to directly assimilate Tb observations using a consistent data assimilation system, but a correct global characterization of the Tb forecast and observation errors remains difficult. The L2 SM retrievals are easily handled products, but their assimilation is impacted by errors introduced by inconsistent ancillary information in the SM retrieval algorithm and the assimilation system. With further improvements in the assimilated retrievals and careful selection of the ancillary data SM retrieval assimilation may become a coequal alternative.

Three different data products from the SMOS mission are assimilated separately into the GEOS-5 land surface model to improve estimates of surface and root-zone soil moisture and to study the workings of each assimilation system. The first product consists of L1-based data of multi-angle, dual-polarization Tb observations at the bottom of the atmosphere. The second product is a derived 40° Tb product that mimics SMAP data. The third product are the operational L2 SM retrievals. Special care is taken during quality control and processing of the satellite observations prior to assimilation and within the

assimilation system. The Tb assimilation uses a distributed EnKF with a temporally variable Tb bias mitigation, a system that is also used for the SMAP L4_SM product (Reichle et al., 2016). The SM retrieval assimilation uses a similar system, but with CDF-matching instead to eliminate the more stationary SM innovation biases. The study covers most of North America for the period of 1 July 2010 - 1 May 2015.

The Tb and SM innovations show very different spatial patterns and the number of assimilated observations differs because of

different needs for data screening and bias mitigation. Based on the average sensitivity of Tb to soil moisture, the magnitude of the Tb innovations is comparably larger than that of the SM innovations, which may either introduce more information or more error into the Tb assimilation system. The Tb and SM retrieval assimilation schemes also yield surprisingly different spatio-temporal increment patterns, leading to very different adjustments to the modeled soil moisture trajectories. Despite these stark differences, the various assimilation schemes yield soil moisture estimates with similar average skill metrics, computed from

a set of 187 SCAN and USCRN sites across the US. Compared to in situ observations, both Tb and SM retrieval assimilation yield anomaly correlations around or larger than 0.6 for both the surface and root-zone soil moisture in 'favorable' areas, where the satellite data are expected to better represent the soil moisture conditions, i.e. in areas with limited topographic complexity and limited vegetation. The anomaly correlation with data assimilation is between 0.5 and 0.6 in non-favorable areas. The data assimilation introduces significant improvements over the model-only simulations for surface soil moisture everywhere, but

the improvement are much larger in favorable areas. For the root zone, improvements are also found, but without statistical significance. While no significant differences in domain-averaged skills can be found between the various assimilation systems, there are large local differences in performance between the Tb and SM retrieval assimilation which may be due to differences in information content and screening of the observations, and differences in how close each of the systems is to an optimal calibration of its model and observation error parameters. Therefore, we expect that soil moisture data assimilation systems



can be further improved only if the systems manage to better simulate the spatial and temporal variations of the actual errors in the model and the observations. Furthermore, the SM retrieval assimilation results will benefit from any future improvement in the SM retrievals.

*Acknowledgements.* The NASA Soil Moisture Active Passive (SMAP) mission supported this study. The NASA Center for Climate Simulation (NCCS) at the Goddard Space Flight Center provided computational resources through the NASA High-End Computing (HEC) Program.



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





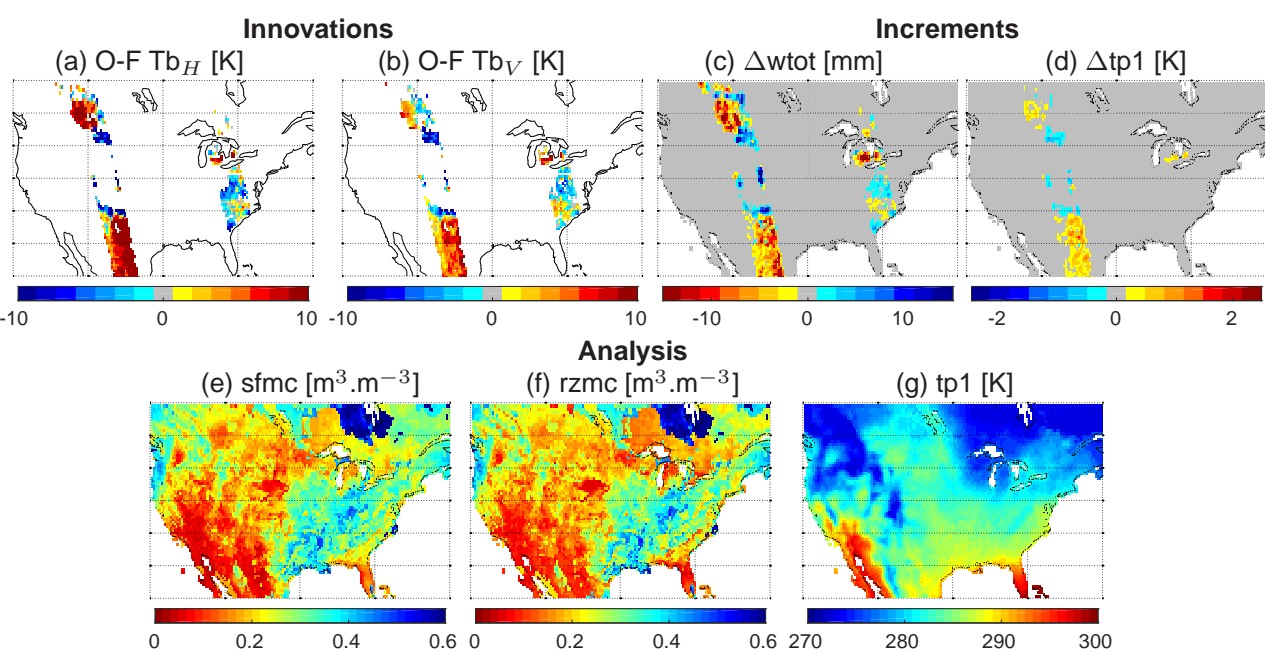

**Figure 1.** Soil moisture and temperature analysis on 30 April 2015 at 12:00 UTC for the Tb_fit assimilation system. (a,b) Tb innovations at 40° incidence angle for H- and V-polarization respectively; (c,d) Increments in total profile water (wtot) and first soil layer temperature (tp1), respectively; (e,f,g) Assimilation analyses of surface soil moisture (sfmc), root-zone soil moisture (rzmc) and soil temperature (tp1), respectively.





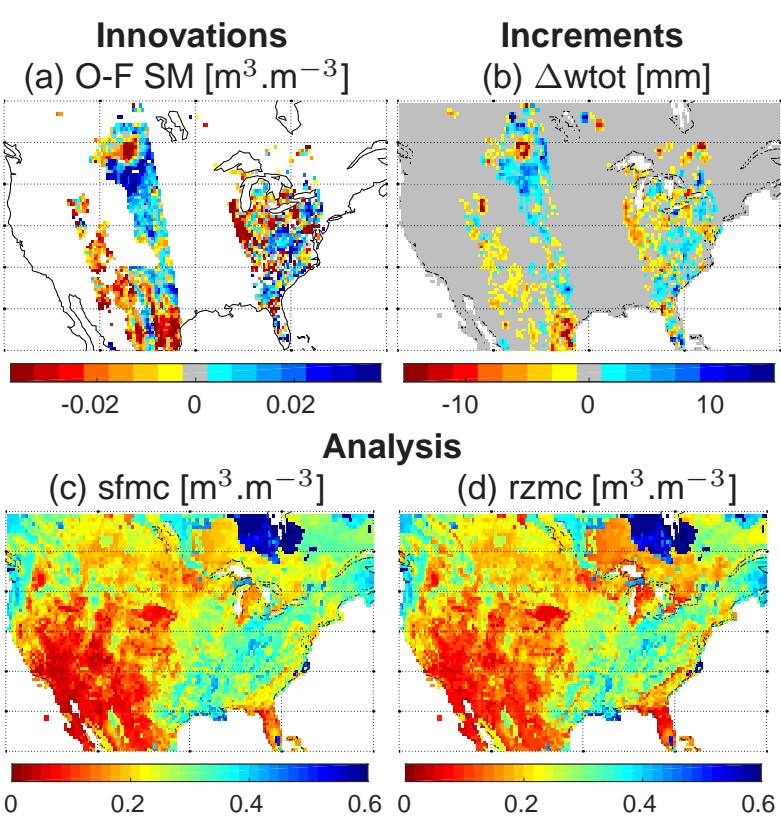

**Figure 2.** Soil moisture analysis on 30 April 2015 at 12:00 UTC for the SM retrieval assimilation system. (a) SM innovations; (b) Increments in total profile water (wtot); (c,d) Assimilation analyses of surface soil moisture (sfmc) and root-zone soil moisture (rzmc).







**Figure 3.** Observation-space assimilation diagnostics for the period from 1 July 2010 to 1 May 2015. Number of assimilated observation sets for (a) Tb_7ang assimilation, (b) Tb_fit assimilation, and (c) SM retrieval assimilation. Standard deviation of the (d) Tb innovations from Tb_7ang assimilation, (e) Tb innovations from Tb_fit assimilation, and (f) SM innovations from SM retrieval assimilation. (g,h,i) same as (d,e,f), but for normalized innovations (normO-F). Ensemble standard deviation of the (g) Tb forecast error for Tb_7ang assimilation, (h) Tb forecast error for Tb_fit assimilation, and (i) surface soil moisture forecast error for SM retrieval assimilation. The titles show the spatial mean (m) and standard deviation (s) across each map.



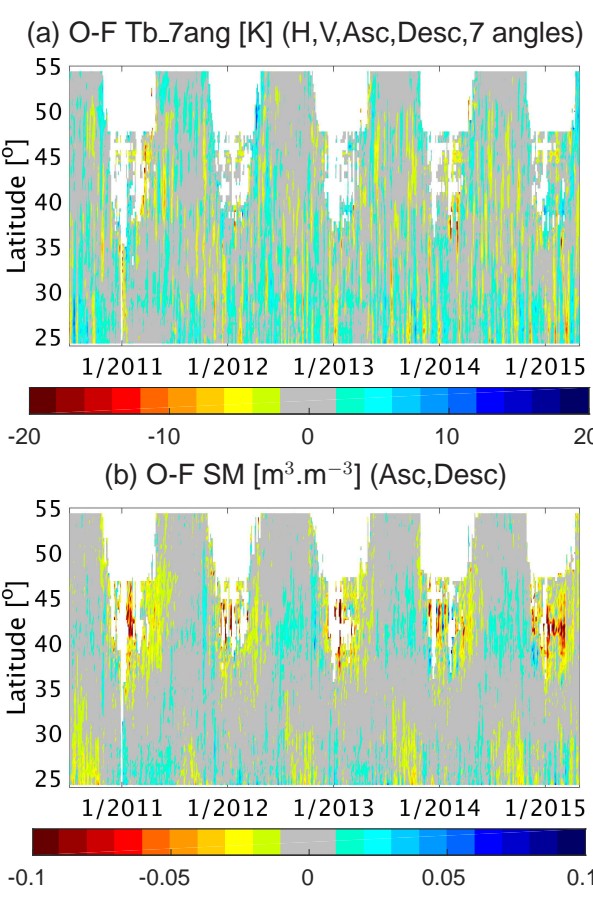

**Figure 4.** Hovmüller plots showing the temporal evolution of longitudinally averaged innovations (O-F) for the period from 1 July 2010 to 1 May 2015. (a) Tb_7ang innovations, averaged over H- and V-polarization, ascending and descending swaths and over 7 incidence angles. (b) SM innovations, averaged over ascending and descending swaths.





**Figure 5.** Statistics of the increments, calculated for the period from 1 July 2010 to 1 May 2015. Number of increments per day for (a) Tb_7ang assimilation, (b) Tb_fit assimilation, and (c) SM assimilation. Temporal standard deviation of total profile water (wtot) increments for (d) Tb_7ang assimilation, (e) Tb_fit assimilation, and (f) SM assimilation. (g,h,i) same as (d,e,f) but for srfexc increments. (j,k,l) same as (d,e,f) but for rzexc increments. (m,n,o) same as (d,e,f) but for catdef increments. The titles show the spatial mean (m) and standard deviation (s) across each map.





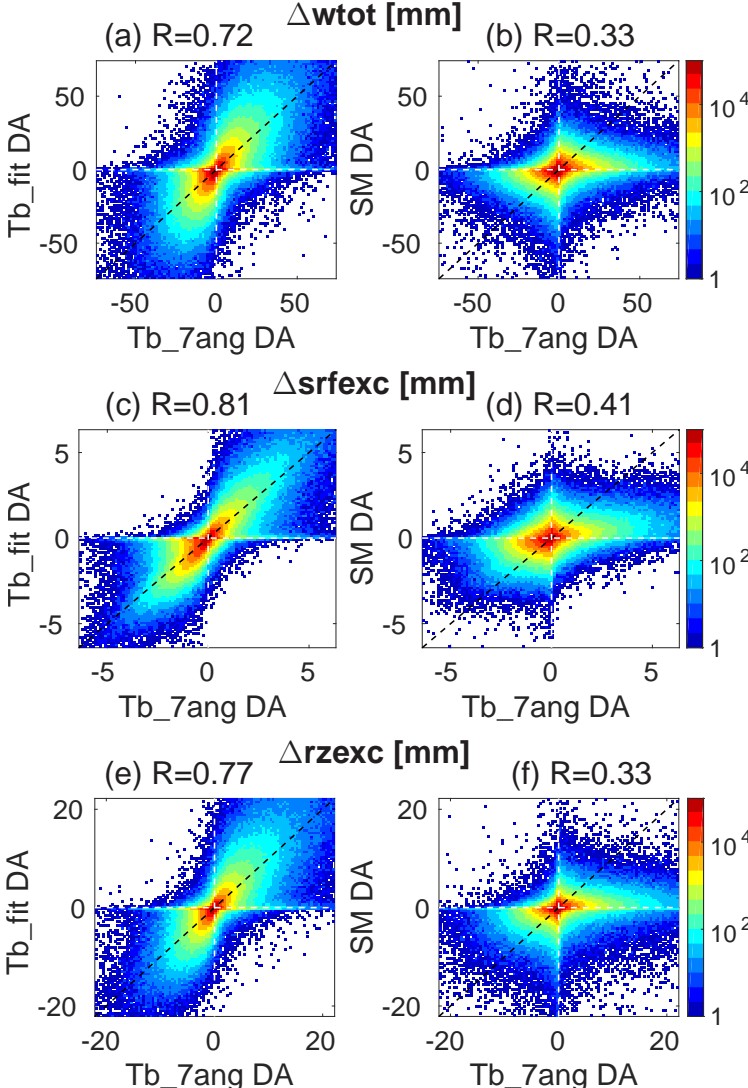

**Figure 6.** Spatially and temporally collocated analysis increments from (a,c,e) Tb_fit assimilation and (b,d,f) SM retrieval assimilation versus same from Tb_7ang assimilation for (a,b) profile-integrated wtot increments, (c,d) srfexc increments, and (e-f) rzexc increments. Increments are from the period 1 July 2010 to 1 May 2015. The plot range is limited to the maximum value of 10 times the standard deviation in either experiment, and divided into 100 even sample bins. Colors indicate the number of sample points within each 1.5 mm bin, 0.13 mm bin or 0.44 mm for $\Delta$wtot, $\Delta$srfexc and $\Delta$rzexc, respectively. R is the spatio-temporal Pearson correlation coefficient between the individual increments from two assimilation experiments.





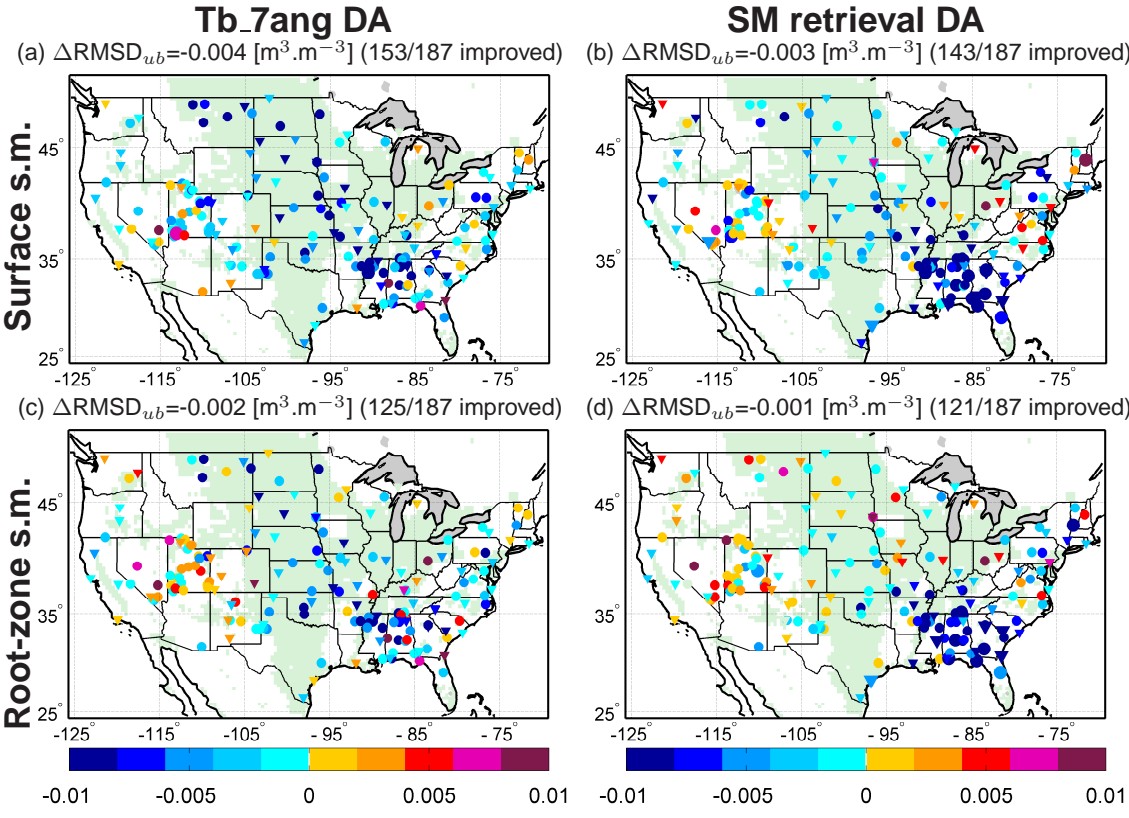

**Figure 7.** Change in unbiased RMSD ($\Delta$RMSD$_{ub}$) due to data assimilation at (circles) SCAN and (triangles) USCRN sites for (a,b) surface and (c,d) root-zone soil moisture, for (a,c) Tb_7ang and (b,d) SM retrieval assimilation. Statistically significant changes are marked by larger symbols. Metrics are calculated across 3-hourly time steps during the period from 1 July 2010 to 1 May 2015. The titles indicate the spatial mean $\Delta$RMSD$_{ub}$ across all sites with clustering (31 clusters). The green background shading marks areas with limited vegetation and topographic complexity based on model parameters.





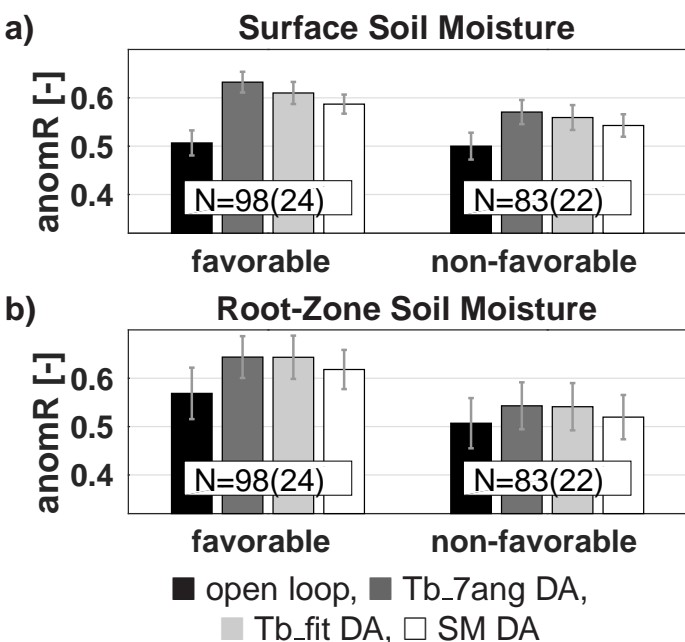

**Figure 8.** Performance of open-loop and data assimilation experiments in terms of anomaly correlations (anomR) calculated across 3-hourly analyses and forecast time steps from 1 July 2010 to 1 May 2015, for (a) surface and (b) root-zone soil moisture. The bars show skill metrics averaged over sites in either favorable or non-favorable areas, where favorable areas refer to the areas indicated by the green background shading in Figure 7. The variable N is the total number of SCAN and USCRN sites considered for each category, with the number of clusters in parentheses. The error bars reflect cluster-averaged 95% confidence intervals.