# Peer review of "Assimilation of SMOS Brightness Temperatures or Soil Moisture Retrievals into a Land Surface Model"

_Hydrology and Earth System Sciences, 2016_

## Referee Comment (RC1) · Anonymous Referee #1 · 19 Sep 2016

General Comments:

This work is trying to assimilate SMOS brightness temperature, soil moisture product and TB difference between SMAP and SMOS into GEOS-5. The outputs are compared with each other, as well as against the SCAN and USCRN in-site soil moisture networks. By using strict mathematical test, inputs quality control and assimilation scheme, the result is reliable and reasonable. The method established in this paper imitates the SMAP Level 4 product but expands its technique. I think it is very useful for the future application of SMOS/SMAP product and further soil moisture-weather/climate model researches. This paper is recommended for publication.

However, I have some doubts, not necessarily comments to the authors:

Line 6, section 3.1, Is the spatially distributed the same concept as three-dimensional? If each grid was updated separately as said in Line 6-7, then which three dimensions are they?

The variables listed in P4, Line 26-30 will be updated after assimilation loop as described in P7, Line 11-15. The variables includes part soil moisture/temperature defined in the land surface model but not all of them. In this case, the soil moisture/temperature will partly altered by the assimilation, is that right? How do the authors select which layer should be assimilated into? Will this selection affect the weather forecasting?

Figure 7 indicates the changes due to assimilation. Large marks are assigned for statistically significant sites but it is really hard to distinguish them from the rest. Maybe the authors could use other symbols or give some explanations on how many sites are improved actually. Besides, what is the evaluation of simulation result without data assimilation while only changes are illustrated in Figure 7? I see the magnitude in changes is quite small, about 0.01, which is lower than the accuracy of SMOS/SMAP mission. If the model simulation doesn't match well (for instance, difference larger than 0.1), how important the improvements brought in by assimilation should be reconsidered. I know the comparison between model and in-site observation is a very complicated issue and may be too much if it is discussed in this paper. What I recommended is to add some simple figures which could give an estimation of model v.s. observation difference, without assimilation.

For Figure 8, the problem still exists. Figure 8 adopts a similar method used in another paper in De Lannoy and Reichle, Journal of Hydrometeorology, 2016. It is for sure that the correlation increase indicates the effect of assimilation but the correlation increase doesn't mean the absolute soil moisture value is improved. Usually in the current forecast model, soil moisture is a diagnostic variable which does not interact with the atmosphere directly. As mentioned in Line 21, if the soil moisture will be used to improve weather forecast, its absolute value is more important for evaporation/Bowen

ratio calculation. By enlarging or restricting soil moisture, the land surface model in climate forecast could also collapse while correlation coefficient increases. Without seeing the soil moisture time series for particular site, or at least any time–series which reflect the variation, the conclusion that assimilation improved soil moisture simulation should be made with caution.

Minors:

P4, Line 3, "][" should be replaced by "[]" P4, Line 10, What is the SM uncertainty? Is it one of the products from SMOS?

---

## Referee Comment (RC2) · Anonymous Referee #2 · 21 Sep 2016

General comments

Thank the authors for this interesting work. Based on a catchment land surface model and the advanced EnKF data assimilation technique, this study employed several experiments trying to answer the question on "how to make the best use of L-band microwave satellite observations through DA". Prior to the assimilation, a sophisticated data quality control, model perturbation, and bias mitigation in both TB and SM retrievals are applied. Finally, DA outputs are carefully evaluated by comparing with in-situ measurements and with special attention on DA innovation and increments. The findings provide important inspirations for further SMAP DA and the manuscript is overall well organized. However, some statements within this manuscript remains unclear

to me and more details are needed. I therefore recommend this manuscript being published in Hydrology and Earth System Sciences by taking care of the following minor comments.

Specific comments

1. P1, Line 9-10: soil moisture evaluations are based on anomaly rather than the absolute values. Thus I would suggest rephrasing this sentence as "... to model-only simulations in terms of unbiased root mean square difference and anomaly correlations during the period ..."

2. P3, Line 5: what does the "treatment" exactly refer to? Do the authors mean RFI and uncertainty screening and regridding as depicted in section 2.2? if yes, I would not consider this as a major difference compared with previous studies.

3. P5, Line 1-6: what is a "footprint scale"? As I read from Table 1 of De Lannoy et al. 2014b, the RTM calibrated parameters are assigned to the same IGBP vegetation class, but how do you manage to make "all 36-km grid cells within one footprint area are assigned the same set of RTM parameters". When you practically do assimilation for a specific "footprint", does all the 36-km grids use the same RTM parameters or they are vegetation-class-dependent? Please elaborate.

4. P5, Line 30: what criterion do you use when excluding frozen soil and snow cover during assimilation?

5. P8, Line 23-25: it looks the representativeness error during the upscaling process as described in P3, Line 26-29, is not considered.

6. P8, Line 6-7: the work of Reichle and Koster (2004, GRL) looks a better reference on CDF-matching. Besides, on what temporal scale did the authors conduct this CDF-matching? Is it for each month, year, or the entire study period (2010-2015)? As also stated by the authors in P10, Line 5-6, could the SM innovation be seasonally corrected as well if do CDF-matching on a monthly basis? Please clarify.

[Figure]

7. P10, Line 13-19: another reason for the degradation of TB assimilation might be the modeled vegetation. Meanwhile, Leaf area index, other than vegetation water content, has been found to be reliable in estimating vegetation optical depth at global scale (Kerr et al. 2012, IEEE-TGRS). To ingest real-time dynamic vegetation observations (e.g., LAI from MODIS) might help mitigate the TB assimilation. In any case, RTM over highly vegetated land cover is always tough, and the authors may consider excluding areas with vegetation water content over 5 kg m-2.

8. The in-situ soil moisture is usually measured at the 5 cm depth whereas the model output represents the first layer's average (0-5 cm), and this vertical depth-mismatch can potentially introduce biases in soil moisture evaluation given that the topsoil moisture usually have larger variations. Horizontally, the direct comparison between model estimate of a gridcell average and point-scale in-situ observations can also be questioned due to high sub-grid heterogeneity. For the former, it could be alleviated by configure the land model to have denser soil layers at the top. For the latter, a way of mitigating this spatial representativeness issue is to compare their spatial averages (e.g., Xia et al. 2014, JH). However, I realize it might be difficult for the authors to reconfigure the land model or redesign the evaluation scheme within this paper but it can be considered in future studies.

9. Similar DA framework has already been used in the SMAP_L4 algorithm to produce a value-added root zone soil moisture. Thus in the conclusion section I would like to see a short paragraph of the authors' speculation on possible improvements in the future SMAP TB and SM assimilation as well as the feasibility of their joint assimilation given that these two products complementarily have different spatial coverage and content different land surface information.

Technical corrections

1. P7, Line 14: "as well as surface soil temperature. . ."

2. P22, Figure 3: should the captions of g, h, and i be for subplots j, k, and l?

Additional references

Reichle, R. H., and R. D. Koster (2004), Bias reduction in short records of satellite soil moisture, Geophysical Research Letters, 31(19), L19501, doi:10.1029/2004GL020938.

Kerr, Y. H., et al. (2012), The SMOS Soil Moisture Retrieval Algorithm, Geoscience and Remote Sensing, IEEE Transactions on, 50(5), 1384-1403, doi:10.1109/TGRS.2012.2184548.

Xia, Y., J. Sheffield, M. Ek, J. Dong, N. Chaney, H. Wei, J. Meng, and E. F. Wood (2014), Evaluation of Multi-Model Simulated Soil Moisture in NLDAS-2, Journal of Hydrology, 512, 107-125, doi:10.1016/j.jhydrol.2014.02.027.

---

## Referee Comment (RC3) · Anonymous Referee #3 · 21 Sep 2016

In order to investigate the impact of the different assimilation schemes (Tb & SM retrieval assimilation) on the skill of surface and root-zone SM estimates, the authors assimilated five years of SMOS Tb data or SM data into the GEOS-5 land surface model and RTM model, using a spatially distributed EnKF. They found that different assimilation systems show surprisingly different spatio-temporal increment patterns, leading to very different adjustments to the modeled soil moisture trajectories. Nevertheless, the various schemes yield SM estimates with similar average skill metrics, introducing significant improvements over the model-only simulations. The manuscript was very well structured. However, considering the complexity of the information delivered, some minor adjustments were needed, for the sake of reader's ease in understanding.

[Figure]

Specific comments: 1. The concept of total soil profile water (Δwtot) was used to investigate the innovations, increments and relevant statistics (e.g. in Figure 1, 2, 5, 6). On the other hand, this Δwtot is a diagnostic variable (e.g. an aggregated variable representing changes in catdef, srfexc, rzexc) and is not a member of the state vector for Tb & SM assimilation. It is not clear if the analysis has been done for catdef, srfexc and rzexc, what is the necessity left (or add value) for doing analysis for Δwtot?

It is suggested to use catdef for Figure 1,2,5 & 6, instead of Delta_wtot. Or, if the authors would like to stick with their choice, a verification/clarification is needed.

2. Paragraph 2 on Page 12 (e.g. between line 4 and line 12). (a) It is said " ... The catdef increments pertain to the entire profile depth ... and have a relatively small impact on the upper 5cm SM ..., would scale to about 0.1 mm for a 5cm soil layer". First of all, catdef is a state variable of CLSM and would exert only second order effects on surface SM (Koster et al. 2000). It is not clear how the authors scaled the catdef increments to surface SM; (b) "The increments in rzexc are relatively smallest, because this variable is not perturbed by design". The rzexc is another state variable of CLSM and was also member of the state vector used by the assimilation system. So, with such context, what does it mean ".. not perturbed by design" here?

3. line 16, page 2, gage-based -> gauged-based;

4. line 17, page 2. Not clear what does it mean here by "inconsistent"? Did you mean here that the ST observed by a different satellite other than the one for SM?

5. line 3, page 4, range between 30 and 50 deg?

6. line 30, page 4. Reichle et al. (2015) is not detailed enough for an overview of the CLSM model variables. Please update them with following references: Koster, R. D., M. J. Suarez, A. Ducharne, M. Stieglitz, and P. Kumar, A catchment-based approach to modeling land surface processes in a GCM, Part 1, Model Structure, J. Geophys. Res., 105, 24809-24822, 2000.

Ducharne, A., R. D. Koster, M. J. Suarez, M. Stieglitz, and P. Kumar, A catchment-based approach to modeling land surface processes in a GCM, Part 2, Parameter estimation and model demonstration, J. Geophys. Res., 105, 24823-24838, 2000.

7.Section 2.3, there are lots of information in this subsection. However, it is difficult to follow without further checking other references. More specifically, for the concept of "footprint scale" and 36-km grid, how they were relevant and how exactly they varied with latitude and longitude are not clear. Perhaps a schematic will help? Furthermore, could the authors help to use a flowchart here to show that RTM converted CLSM simulations into Tb and then this calculated Tb was used to compute O-F residuals inthe assimilation system, while considering the geometric relations between "footprint scale" and "36km grid"? Such flowchart will help tremendously the readers to have an overview of the whole processing chain, which is the most fundamental for understanding the topic of this manuscript.

8. line 26, page 5, how were the weights assigned, with different depths?

9. line 1-2, page 6, any reference for this statement?

10. section 3.1, again, this is also fundamental for readers to understand the manuscript. Could the authors help to add a flowchart here to show the difference between the Tb and SM assimilation algorithm?

---

## Author Comment (AC3) · 17 Oct 2016

*We thank all reviewers for their suggestions. The original comments are in black normal fonts. The answers are in blue italic fonts. Modified text is underlined. Figure, page and line numbers generally refer to the old manuscript. An indication of anticipated new figure, page and line numbers are provided in brackets.*

Referee #3

In order to investigate the impact of the different assimilation schemes (Tb & SM re- trieval assimilation) on the skill of surface and root-zone SM estimates, the authors as- similated five years of SMOS Tb data or SM data into the GEOS-5 land surface model and RTM model, using a spatially distributed EnKF. They found that different assimilation systems show surprisingly different spatio-temporal increment patterns, leading to very different adjustments to the modeled soil moisture trajectories. Nevertheless, the various schemes yield SM estimates with similar average skill metrics, introducing sig- nificant improvements over the model-only simulations. The manuscript was very well structured. However, considering the complexity of the information delivered, some minor adjustments were needed, for the sake of reader's ease in understanding.

*We thank the reviewer for the encouraging review and for all suggestions.*

Specific comments: 1. The concept of total soil profile water ($\Delta$wtot) was used to investigate the innovations, increments and relevant statistics (e.g. in Figure 1, 2, 5, 6). On the other hand, this $\Delta$wtot is a diagnostic variable (e.g. an aggregated variable representing changes in catdef, srfexc, rzexc) and is not a member of the state vector for Tb & SM assimilation. It is not clear if the analysis has been done for catdef, srfexc and rzexc, what is the necessity left (or add value) for doing analysis for $\Delta$wtot?

It is suggested to use catdef for Figure 1,2,5 & 6, instead of Delta_wtot. Or, if the authors would like to stick with their choice, a verification/clarification is needed.

*We agree to edit the text for clarification (p.7, L6 [p.7, L27]).*

2. Paragraph 2 on Page 12 (e.g. between line 4 and line 12). (a) It is said " ... The catdef increments pertain to the entire profile depth ... and have a relatively small impact on the upper 5cm SM ..., would scale to about 0.1 mm for a 5cm soil layer". First of all, catdef is a state variable of CLSM and would exert only second order effects on surface SM (Koster et al. 2000). It is not clear how the authors scaled the catdef increments to surface SM; (b) "The increments in rzexc are relatively smallest, because this variable is not perturbed by design". The rzexc is another state variable of CLSM and was also member of the state vector used by the assimilation system. So, with such context, what does it mean ".. not perturbed by design" here?

*We will further clarify the details (p.12, L10 [p.12, L30]).*

3. line 16, page 2, gage-based -> gauged-based;

*We will edit this to read gauge throughout the text.*

4. line 17, page 2. Not clear what does it mean here by "inconsistent"? Did you mean here that the ST observed by a different satellite other than the one for SM?

*ST is estimated by a modeling system, not even by a satellite. We will rephrase this for clarification (p.2, L17 [p.2, L18]).*

5. line 3, page 4, range between 30 and 50 deg?

*Sure, we will edit this (p.4, L3, [p.4, L6]).*

6. line 30, page 4. Reichle et al. (2015) is not detailed enough for an overview of the CLSM model variables. Please update them with following references: Koster, R. D., M. J. Suarez, A. Ducharne, M. Stieglitz, and P. Kumar, A catchment-based approach to modeling land surface processes in a GCM, Part 1, Model Structure, J. Geophys. Res., 105, 24809-24822, 2000. Ducharne, A., R. D. Koster, M. J. Suarez, M. Stieglitz, and P. Kumar, A catchment- based approach to modeling land surface processes in a GCM, Part 2, Parameter estimation and model demonstration, J. Geophys. Res., 105, 24823-24838, 2000.

*We will add the suggested references (p.4, L30 [p.5, L2]).*

7. Section 2.3, there are lots of information in this subsection. However, it is difficult to follow without further checking other references. More specifically, for the concept of "footprint scale" and 36-km grid, how they were relevant and how exactly they varied with latitude and longitude are not clear. Perhaps a schematic will help? Furthermore, could the authors help to use a flowchart here to show that RTM converted CLSM simulations into Tb and then this calculated Tb was used to compute O-F residuals inthe assimilation system, while considering the geometric relations between "footprint scale" and "36km grid"? Such flowchart will help tremendously the readers to have an overview of the whole processing chain, which is the most fundamental for understanding the topic of this manuscript.

*We will add a schematic overview of the assimilation algorithm with special attention to the resolutions for clarification (p.6, L22 [p.7, L16]).*

8. line 26, page 5, how were the weights assigned, with different depths?

*This will be edited (p.5, L27 [p.6, L2]).*

9. line 1-2, page 6, any reference for this statement?

*We will add a reference (p.6, L2 [p.6, L14])*

10. section 3.1, again, this is also fundamental for readers to understand the manuscript. Could the authors help to add a flowchart here to show the difference between the Tb and SM assimilation algorithm?

*See above, comment 7. We will add a flowchart.*

---

## Author Response (AR1)

*We thank all reviewers for their suggestions. The original comments are in black normal fonts. The answers are in blue italic fonts. Modified text is underlined. Figure, page and line numbers generally refer to the old manuscript. New page and line numbers are provided in brackets.*

Referee #1

General Comments:

This work is trying to assimilate SMOS brightness temperature, soil moisture product and TB difference between SMAP and SMOS into GEOS-5. The outputs are compared with each other, as well as against the SCAN and USCRN in-site soil moisture networks. By using strict mathematical test, inputs quality control and assimilation scheme, the result is reliable and reasonable. The method established in this paper imitates the SMAP Level 4 product but expands its technique. I think it is very useful for the future application of SMOS/SMAP product and further soil moisture-weather/climate model researches. This paper is recommended for publication.

*We thank the reviewer for the encouraging review and for all suggestions.*

However, I have some doubts, not necessarily comments to the authors:

1) Line 6, section 3.1, Is the spatially distributed the same concept as three-dimensional? If each grid was updated separately as said in Line 6-7, then which three dimensions are they?

*Indeed, "spatially distributed" and "three-dimensional" refers to the same concept as was already noted in the text. The 3 dimensions are in space. This is edited as follows:*

*(p.5, L7 [p.6, L20]) "This system simultaneously assimilates multiple spatially distributed observation sets, using horizontal and vertical error covariance structures, to update the simulations at each 36-km model grid cell."*

2) The variables listed in P4, Line 26-30 will be updated after assimilation loop as described in P7, Line 11-15. The variables includes part soil moisture/temperature defined in the land surface model but not all of them. In this case, the soil moisture/temperature will partly altered by the assimilation, is that right? How do the authors select which layer should be assimilated into? Will this selection affect the weather forecasting?

*Indeed, the deeper layer soil temperature is not included in the Kalman filter update, but it will receive indirect updates though downward propagation of the information by the soil heat diffusion module of the Catchment model. The observations are most directly related to the surface layer (5 cm) soil moisture, but the observations are used to update all soil moisture variables related to the entire soil profile. This is edited as follows:*

*(p.7, L4 [p.7, L21]) "The state vector for Tb assimilation ….includes prognostic variables related to soil moisture and soil temperature, because Tb observations are by definition sensitive to surface soil moisture and temperature. In contrast, the state vector for SM retrieval assimilation … contains only model prognostic variables related to soil moisture, because the SM retrievals do not carry direct information about the soil temperature. The select updates will be propagated to all other variables within the land surface modeling system through energy and water exchange between various soil layers and land-vegetation-atmosphere compartments."*

*The modeling and assimilation system used in the study includes only the land (section 2.3), and impacts on weather forecasting are beyond the scope of the present paper. But we anticipate that weather forecasts would primarily be impacted by the updated soil moisture (including surface and root zone) via evapotranspiration and by the surface temperature via sensible heat. The lower-layer soil temperatures should have at most a second-order impact.*

3) Figure 7 indicates the changes due to assimilation. Large marks are assigned for statistically significant sites but it is really hard to distinguish them from the rest. Maybe the authors could use other symbols or give some explanations on how many sites are improved actually. Besides, what is the evaluation of simulation result without data assimilation while only changes are illustrated in Figure 7? I see the magnitude in changes is quite small, about 0.01, which is lower than the accuracy of SMOS/SMAP mission. If the model simulation doesn't match well (for instance, difference larger than 0.1), how important the improvements brought in by assimilation should be re- considered. I know the comparison between model and in-site observation is a very complicated issue and may be too much if it is discussed in this paper. What I rec- ommended is to add some simple figures which could give an estimation of model v.s. observation difference, without assimilation.

*We updated the old Figure 7 ([Figure 8]) to also include the RMSD_ub values for the open loop simulation, as suggested by the reviewer:*

*(p.14, L3 [p.14, L13]) "Figure 8 shows the RMSDub (section 2.4) for the model-only open loop (OL) simulation, and the change in RMSDub…"*

*[p.14, L17] "The OL simulation has an average RMSDub value of 0.054 $m^3.m^{-3}$ for surface soil moisture and 0.039 $m^3.m^{-3}$ for root-zone soil moisture. Looking more closely, the RMSDub values are generally higher in the central and wetter eastern regions. In dry areas, the RMSDub is limited, because the time series show a limited variability for lack of much precipitation…."*

*[p.14, L25] "The domain-average ΔRMSDub values caused by assimilation are only barely statistically significant for surface soil moisture in `favorable' areas, i.e. where the satellite observations are most sensitive to soil moisture (indicated with green background shading in Figure 8). The differences between Tb_7ang, Tb_fit or SM retrieval assimilation are not significant. The assimilation contributes an average relative improvement in*

*surface soil moisture of 7% of the OL RMSDub in favorable locations and 4% in non-favorable areas. Both Tb and SM retrieval assimilation show improvements in the central and eastern parts of the US, but perform poorly in the western dry mountain areas, where the RMSDub for the OL was small and the assimilation may have introduced some additional noise. The Tb_7ang assimilation shows the largest improvements in the central US, whereas the SM retrieval assimilation shows the largest improvements in the southeastern part, for both surface and root-zone soil moisture...."*

[Figure]

*"Figure 8: Unbiased RMSD (RMSDub) for the model-only open loop (OL) simulation, and change in unbiased RMSD (ΔRMSDub) due to data assimilation at (circles) SCAN and (triangles) USCRN sites for (a,b,c) surface and (d,e,f) root-zone soil moisture. The skill of (a,d) the open loop simulation is the reference value for the changes in skill due to (b,e) Tb_7ang and (c,f) SM retrieval assimilation. Statistically significant changes are marked by larger symbols (e.g. southeastern US for SM retrieval assimilation). Metrics are calculated across 3-hourly time steps during the period from 1 July 2010 to 1 May 2015. The titles indicate the spatial mean (Δ)RMSDub across all sites with clustering (31 clusters). The gray background shading marks areas with limited vegetation and topographic complexity based on model parameters."*

4) For Figure 8, the problem still exists. Figure 8 adopts a similar method used in an- other paper in De Lannoy and Reichle, Journal of Hydrometeorology, 2016. It is for sure that the correlation increase indicates the effect of assimilation but the correlation increase doesn't mean the absolute soil moisture value is improved. Usually in the cur- rent forecast model, soil moisture is a diagnostic variable which does not interact with the atmosphere directly. As mentioned in Line 21 *(p.1)*, if the soil moisture will be used to improve weather forecast, its absolute value is more important for evaporation/Bowen ratio calculation. By enlarging or restricting soil moisture, the land surface model in climate forecast could also collapse while correlation coefficient increases. Without seeing the soil moisture time series for particular site, or at least any time–series which reflect the variation, the conclusion that assimilation improved soil moisture simulation should be made with caution.

*It is not 100% clear to us what the reviewer means by improved "absolute soil moisture value." Our assimilation system is primarily designed to estimate or improve temporal variations of soil moisture by correcting for random errors. The long-term mean soil moisture values from the assimilation are intentionally the same as those of the underlying model. (This implies that the assimilation estimates can still be used in conjunction with the **model's** relationship between soil moisture and the Bowen ratio, without upsetting the forecast model's calibration.) In any case, when the model soil moisture is adjusted by the analysis updates, its absolute value changes at any given time. The correlation and $RMSD_{ub}$ metrics do measure this change (but would not measure a change in the long-term mean, which, again, does not occur by design).*

*We edited this as follows:*

*(p.5 [p.5, L32]) "2.4 In Situ Soil Moisture Data and Metrics"*

*(p.5, L30 [p.6, L7-9]) "....the skill is quantified in terms of anomaly time series correlation (anomR),.... The anomaly correlation is based on anomaly time series obtained by subtracting a multi-year smoothed climatology from both the simulations and in situ observations. Note that the assimilation and open-loop simulations have, by design, the same climatological variability; the assimilation only corrects for random errors."*

Minors:

P4, Line 3, "][" should be replaced by "[]" P4, Line 10, What is the SM uncertainty? Is it one of the products from SMOS?

*- In fact, the "]..[" is correct, b/c we exclude 30º and 50º incidence angles from the interval. Since reviewer#3 also stumbled over it, we now spelled it out.*

*(p.4, L3 [p.4, L6]) "…10 data points in the incidence angle interval between 30º and 50º."*

*- The SM retrieval uncertainty is provided inside the SMOS product. This is clarified:*

*(p.4, L9 [p.4, L12]): "Based on the quality information provided within the SMOS products, the SM are retained only if:…."*

Referee #2

General comments

Thank the authors for this interesting work. Based on a catchment land surface model and the advanced EnKF data assimilation technique, this study employed several ex- periments trying to answer the question on "how to make the best use of L-band mi- crowave satellite observations through DA". Prior to the assimilation, a sophisticated data quality control, model perturbation, and bias mitigation in both TB and SM retrievals are applied. Finally, DA outputs are carefully evaluated by comparing with in-situ measurements and with special attention on DA innovation and increments. The findings provide important inspirations for further SMAP DA and the manuscript is overall well organized. However, some statements within this manuscript remains unclear to me and more details are needed. I therefore recommend this manuscript being published in Hydrology and Earth System Sciences by taking care of the following minor comments.

*We thank the reviewer for the encouraging review and for all suggestions.*

Specific comments

1. P1, Line 9-10: soil moisture evaluations are based on anomaly rather than the absolute values. Thus I would suggest rephrasing this sentence as ". . . to model-only simulations in terms of unbiased root mean square difference and anomaly correlations during the period . . ."

*This is edited accordingly*

*(p.1, L9-10 [p.1, L9-10]) "All assimilation experiments improve the soil moisture estimates compared to model-only simulations in terms of unbiased root-mean-square differences and anomaly correlations during the period 1 July 2010 to 1 May 2015 and for 187 sites across the United States."*

2. P3, Line 5: what does the "treatment" exactly refer to? Do the authors mean RFI and uncertainty screening and re-gridding as depicted in section 2.2? if yes, I would not consider this as a major difference compared with previous studies.

*The "treatment" mainly refers to the interpretation of the spatial support of the SMOS retrievals. For the Tb observations and observation prediction, the "treatment" refers to both the spatial support and the improved RTM forward simulation. This is edited for clarification:*

*(p.2, L28 [p.2, L27]) "...and an improved spatial support and forward simulation of the Tb observation predictions."*

*(p.3 L5 [p.3, L7])"… in the advanced quality screening and spatial support of the SM retrieval observations…"*

*And additionally:*

*(p.5, L6 [p.5, L13-15]) "The methodology is very similar to that in De Lannoy and Reichle, 2016, but with the difference that, here, the RTM does not simulate atmospheric contributions (because the Tb observations are now a priori corrected for atmospheric contributions) and the observation predictions are spatially aggregated using a realistic (but approximate) antenna pattern."*

3. P5, Line 1-6: what is a "footprint scale"? As I read from Table 1 of De Lannoy et al. 2014b, the RTM calibrated parameters are assigned to the same IGBP vegetation class, but how do you manage to make "all 36-km grid cells within one footprint area are assigned the same set of RTM parameters". When you practically do assimilation for a specific "footprint", does all the 36-km grids use the same RTM parameters or they are vegetation-class-dependent? Please elaborate.

*The footprint scale refers to the area effectively observed by the satellite. We rephrased the statement and further elaborate in the text to explain that each 36-km pixel uses its own RTM parameters in forward modeling of the data assimilation. With regard to Table 1 in De Lannoy et al. (2014b): there is a little misunderstanding here, as this table showed the imposed background information (not calibrated) per vegetation class, whereas the calibration definitely happens for each pixel separately.*

*(p.5, L2 [p.5, L8-11]) "…all 36-km grid cells within one footprint area are initially assigned the same set of RTM parameters… For each 36-km grid cell, the calibration estimates a spatially homogeneous set of RTM parameters for the entire associated footprint area and the resulting values are assigned to the central (and dominant) 36-km grid cell only. For the forward calculation of the Tb observation predictions during the data assimilation, all 36-km pixels have a unique set of RTM parameters."*

4. P5, Line 30: what criterion do you use when excluding frozen soil and snow cover during assimilation?

*We edited the text to explicitly include the thresholds:*

*(p.5, L30 [p.6, L6]) "excluding times when the soil is frozen (top layer soil temperature < 274.15 K) or snow covered (snow water equivalent > 0 kg m$^{-2}$)."*

5. P8, Line 23-25: it looks the representativeness error during the upscaling process as described in P3, Line 26-29, is not considered.

*We agree that the text needs a clarification:*

*(p.5, L25 [p.9, L10]) "...due to errors in the RTM, the spatial aggregation, or other discrepancies…"*

6. P8, Line 6-7: the work of Reichle and Koster (2004, GRL) looks a better reference on CDF-matching. Besides, on what temporal scale did the authors conduct this CDF-matching? Is it for each month, year, or the entire study period (2010-2015)? As also stated by the authors in P10, Line 5-6, could the SM innovation be seasonally corrected as well if do CDF-matching on a monthly basis? Please clarify.

*We agree; this was a bibtex/latex glitch. The reference is updated.*

*The text already mentioned that the retrievals were not corrected seasonally, but we rephrased it slightly to further clarify the issue:*

*(p.8, L8 [p.8, L25-28]) "Therefore, the SM innovation biases are not corrected seasonally, but instead cumulative distribution function (CDF) matching between the observations and simulations is performed (Reichle and Koster, 2004) to reconcile the differences in long-term mean, variance and higher moments, as in earlier retrieval assimilation studies (Liu et al., 2011; Draper et al., 2012). The observed and simulated SM CDFs are computed for the entire study period 1 July 2010 - 1 May 2015 at each 36-km grid cell individually..."*

7. P10, Line 13-19: another reason for the degradation of TB assimilation might be the modeled vegetation. Meanwhile, Leaf area index, other than vegetation water content, has been found to be reliable in estimating vegetation optical depth at global scale (Kerr et al. 2012, IEEE-TGRS). To ingest real-time dynamic vegetation observations (e.g., LAI from MODIS) might help mitigate the TB assimilation. In any case, RTM over highly vegetated land cover is always tough, and the authors may consider excluding areas with vegetation water content over 5 kg m-2.

*We agree with this thought and we believe that it was already articulated in our submitted manuscript, but we added an extra note for clarification. We already had stratified the data assimilation results into favorable and non-favorable areas (Fig. 7-8), with favorable areas excluding grid cells with vegetation water content greater than 5 kg/m$^2$.*

*(p.10, L18 [p.11, L1]) "...with larger values (exceeding 10 K) in the central plains and along the Mississippi, where agricultural practices, such as altering crop rotation and irrigation, are observed by SMOS, whereas interannual variations in vegetation are not simulated by the model or provided as input to the model."*

8. The in-situ soil moisture is usually measured at the 5 cm depth whereas the model output represents the first layer's average (0-5 cm), and this vertical depth-mismatch can potentially introduce biases in soil moisture evaluation given that the topsoil moisture usually have larger variations. Horizontally, the direct comparison between model estimate of a gridcell average and point-scale in-situ observations can also be ques- tioned due to high sub-grid heterogeneity. For the former, it could be alleviated by configure the land model to have denser soil layers at the top. For the latter, a way of mitigating this spatial

representativeness issue is to compare their spatial averages (e.g., Xia et al. 2014, JH). However, I realize it might be difficult for the authors to reconfigure the land model or redesign the evaluation scheme within this paper but it can be considered in future studies.

*We agree and we will keep this in mind, but indeed the model cannot be easily re-configured, because it is part of a larger operational system.*

*The issue of the spatial (vertical and horizontal) mismatch between in situ measurements and model or assimilation results is partly circumvented in our paper by using bias-free validation metrics, as clarified on p.5, L.27 [p.6, L4]. Also, we prefer to first calculate the metrics at each site and then take a spatial average, instead of switching these two operations, because we believe that our sequence of operations is more conservative and has less risk of hiding local problems.*

9. Similar DA framework has already been used in the SMAP_L4 algorithm to produce a value-added root zone soil moisture. Thus in the conclusion section I would like to see a short paragraph of the authors' speculation on possible improvements in the future SMAP TB and SM assimilation as well as the feasibility of their joint assimilation given that these two products complementarily have different spatial coverage and content different land surface information.

*Our reference to Reichle et al., 2016 gives the latest available update on the SMAP L4 product. We further added a few thoughts in line with the suggestion by the reviewer:*

*([p.17, last paragraph]) "In line with our findings for the SMOS data assimilation, we anticipate that future versions of the SMAP Tb assimilation system for the L4_SM product may benefit from an improved characterization of spatial model and observation error structures, and from a better representation of some modeling components, such as e.g. vegetation. In addition, given that SMOS and SMAP both provide L-band Tb observations, future assimilation systems should consider a joint assimilation of SMOS and SMAP Tb data. In such a system, it is important to consider the different instrument, Tb processing and Tb error characteristics of the two L-band missions (De Lannoy et al., 2015)."*

Technical corrections

1. P7, Line 14: "as well as surface soil temperature. . ."

*The original sentence is actually correct, but for clarification we adjusted it to:*

*(p.7, L14 [p.7, L35]) "... complete analysis maps of surface and root-zone soil moisture, as well as surface temperature and soil temperature".*

2. P22, Figure 3: should the captions of g, h, and i be for subplots j, k, and l?

*(Figure 3 [Figure 4]) Good catch, many thanks.*

Additional references

Reichle, R. H., and R. D. Koster (2004), Bias reduction in short records of satellite soil moisture, Geophysical Research Letters, 31(19), L19501, doi:10.1029/2004GL020938.

Kerr, Y. H., et al. (2012), The SMOS Soil Moisture Retrieval Algorithm, Geoscience and Remote Sensing, IEEE Transactions on, 50(5), 1384-1403, doi:10.1109/TGRS.2012.2184548.

Xia, Y., J. Sheffield, M. Ek, J. Dong, N. Chaney, H. Wei, J. Meng, and E. F. Wood (2014), Evaluation of Multi-Model Simulated Soil Moisture in NLDAS-2, Journal of Hydrology, 512, 107-125, doi:10.1016/j.jhydrol.2014.02.027.

In order to investigate the impact of the different assimilation schemes (Tb & SM re- trieval assimilation) on the skill of surface and root-zone SM estimates, the authors as- similated five years of SMOS Tb data or SM data into the GEOS-5 land surface model and RTM model, using a spatially distributed EnKF. They found that different assimila- tion systems show surprisingly different spatio-temporal increment patterns, leading to very different adjustments to the modeled soil moisture trajectories. Nevertheless, the various schemes yield SM estimates with similar average skill metrics, introducing sig- nificant improvements over the model-only simulations. The manuscript was very well structured. However, considering the complexity of the information delivered, some minor adjustments were needed, for the sake of reader's ease in understanding.

*We thank the reviewer for the encouraging review and for all suggestions.*

Specific comments: 1. The concept of total soil profile water (Δwtot) was used to investigate the innovations, increments and relevant statistics (e.g. in Figure 1, 2, 5, 6). On the other hand, this Δwtot is a diagnostic variable (e.g. an aggregated variable representing changes in catdef, srfexc, rzexc) and is not a member of the state vector for Tb & SM assimilation. It is not clear if the analysis has been done for catdef, srfexc and rzexc, what is the necessity left (or add value) for doing analysis for Δwtot?

It is suggested to use catdef for Figure 1,2,5 & 6, instead of Delta_wtot. Or, if the authors would like to stick with their choice, a verification/clarification is needed.

*We agree to edit the text for clarification:*

*(p.7, L6 [p.7, L26]) "For the discussion of the soil moisture increments we will focus on the total profile water increments (Δwtot=Δsrfexc+Δrzexc-Δcatdef) in units of kg m-2 (or, mm of water equivalent). This quantity is easily understandable and thus simplifies the discussion. "*

2. Paragraph 2 on Page 12 (e.g. between line 4 and line 12). (a) It is said " ... The catdef increments pertain to the entire profile depth ... and have a relatively small impact on the upper 5cm SM ..., would scale to about 0.1 mm for a 5cm soil layer". First of all, catdef is a state variable of CLSM and would exert only second order effects on surface SM (Koster et al. 2000). It is not clear how the authors scaled the catdef increments to surface SM; (b) "The increments in rzexc are relatively smallest, because this variable is not perturbed by design". The rzexc is another state variable of CLSM and was also member of the state vector used by the assimilation system. So, with such context, what does it mean ".. not perturbed by design" here?

*We further clarified the details as follows:*

*(p.12, L10 [p.12, L30]) "The catdef increments pertain to the entire profile depth (which varies, with typical values around 2-3~m) and they presumably have a relatively small*

*impact on the upper 5~cm soil layer: the domain-averaged magnitude of 5.4~mm, 4.9~mm and 3.5~mm… would linearly scale to about 0.1~mm for a 5~cm soil layer. This is a rough approximation: in reality the part of catdef that contributes to the 5~cm soil moisture cannot be calculated without computing the entire balanced profile. Yet, the approximate 0.1~mm is considerably less than the 0.6, 0.4 and 0.4~mm for the corresponding srfexc increments."*

3. line 16, page 2, gage-based -> gauged-based;

*We edited this to read gauge throughout the text.*

4. line 17, page 2. Not clear what does it mean here by "inconsistent"? Did you mean here that the ST observed by a different satellite other than the one for SM?

*ST is estimated by a modeling system, not even by a satellite. We moved and rephrased this for clarification:*

*(p.2, L17 [p.2, L30]) "A key disadvantage of a system that assimilates SM retrievals is that the SM retrievals may be produced with inconsistent ancillary data, such as for example soil temperature simulated by another model than that used in the assimilation system. The current SMOS SM retrievals by themselves have been found…"*

5. line 3, page 4, range between 30 and 50 deg?

*Sure, we edited this:*

*(p.4, L3, [p.4, L6]) "…10 data points in the incidence angle interval between 30º and 50º."*

6. line 30, page 4. Reichle et al. (2015) is not detailed enough for an overview of the CLSM model variables. Please update them with following references: Koster, R. D., M. J. Suarez, A. Ducharne, M. Stieglitz, and P. Kumar, A catchment-based approach to modeling land surface processes in a GCM, Part 1, Model Structure, J. Geophys. Res., 105, 24809-24822, 2000. Ducharne, A., R. D. Koster, M. J. Suarez, M. Stieglitz, and P. Kumar, A catchment-based approach to modeling land surface processes in a GCM, Part 2, Parameter estimation and model demonstration, J. Geophys. Res., 105, 24823-24838, 2000.

*We added the suggested references.*

*(p.4, L30 [p.5, L2]) "An overview of the model variables is given in Reichle et al. (2015), Koster et al. (2000), and Ducharne et al. (2000)."*

7.Section 2.3, there are lots of information in this subsection. However, it is difficult to follow without further checking other references. More specifically, for the concept of "footprint scale" and 36-km grid, how they were relevant and how exactly they varied with latitude and longitude are not clear. Perhaps a schematic will help? Furthermore, could the authors help to use a flowchart here to show that RTM converted CLSM simulations into

Tb and then this calculated Tb was used to compute O-F residuals inthe assimilation system, while considering the geometric relations between "footprint scale" and "36km grid"? Such flowchart will help tremendously the readers to have an overview of the whole processing chain, which is the most fundamental for understanding the topic of this manuscript.

*We added a schematic overview of the assimilation algorithm with special attention to the resolutions for clarification:*

*[p.5, L19]: "(O-F) residuals in the assimilation system (section 3.1, Figure 1)…"*

*(p.6, L22 [p.7, L16]) "Figure 1 illustrates the forward simulation from 36-km gridded land surface simulations to footprint-scale observation predictions of Tb, and the downscaling of the footprint-scale Tb innovations to 36-km gridded land surface increments."*

[Figure]

*Figure 1: Flowchart of Tb assimilation. The forward simulation consists of (a) land surface model simulations and (b) Tb simulations on the 36-km EASEv2 grid. The Tb simulations are subsequently (c) aggregated using weights based on an approximate antenna pattern. The resulting footprint-scale brightness temperature observation predictions are compared to (d) SMOS observations to calculate innovations (O-F) at the footprint scale. (e) The 3D EnKF maps the footprint-scale innovations to the 36-km EASEv2 grid based on the modeled error correlations between the footprint-scale Tb and the 36-km soil moisture and soil temperature state variables (per Equations 1 and 2).*

8. line 26, page 5, how were the weights assigned, with different depths?

*This is edited as*

*(p.5, L27 [p.6, L2]) "... of measurements at 5, 10, 20, and 50 cm depth, with respective weights of 0.1, 0.1, 0.27 and 0.53."*

9. line 1-2, page 6, any reference for this statement?

*We added a reference:*

*(p.6, L2 [p.6, L14]) "cluster radius of 3º, which approximately reflects the autocorrelation length of large-scale topographic and meteorological phenomena, or of large-scale soil moisture patterns (Vinnikov et al., 1996)."*

10. section 3.1, again, this is also fundamental for readers to understand the manuscript. Could the authors help to add a flowchart here to show the difference between the Tb and SM assimilation algorithm?

*See above, comment 7. We added a flowchart.*

[revised manuscript text omitted]